



# An empirical method for absolute calibration of coccolith thickness

Saúl González-Lemos[1*], José Guitián[2*], Miguel-Ángel Fuertes[3], José-Abel Flores[3], Heather M. Stoll[1,2]

[1]Department of Geology, University of Oviedo, Oviedo, 33005, Spain
[2]Department of Earth Sciences, ETH Zurich, Zurich, 8092, Switzerland
[3]Department of Geology, University of Salamanca, Salamanca, 37008, Spain
*These authors contributed equally to this work

*Correspondence   to*:   Saúl   González-Lemos   (saulgonzalezlemos@gmail.com)   and   Heather   M.   Stoll
(heather.stoll@erdw.ethz.ch)

**Abstract.** As major calcifiers in the open ocean, coccolithophores play a key role in the marine carbon cycle. Because they
may be sensitive to changing $CO_2$ and ocean acidification, there is significant interest in quantifying past and present
variations in their cellular calcification by quantifying the thickness of the coccoliths or calcite plates that cover their cells.
Polarized light microscopy has emerged as a key tool for quantifying the thickness of these calcite plates, but the
reproducibility and accuracy of such determinations has been limited by the absence of suitable calibration materials in the
thickness range of coccoliths (0-4 microns). Here, we describe the fabrication of a calcite wedge with a constant slope over
this thickness range, and the independent determination of calcite thickness along the wedge profile. We show how the
calcite wedge provides more robust calibrations in the 0 to 1.55 μm range than previous approaches using rhabdoliths. We
show the particular advantages of the calcite wedge approach for developing equations to relate thickness to the interference
colors that arise in calcite in the thickness range between 1.55 and 4 μm. The calcite wedge approach can be applied to
develop equations relevant to the particular light spectra and intensity of any polarized light microscope system and could
significantly improve within and inter-laboratory data comparability.

## 1 Introduction

The calcification by coccolithophorid algae represents a major source of $CaCO_3$ production in the open oceans (Siesser,
1993). This calcite is believed to be key to ocean biogeochemical cycles, as it increases deep export of organic carbon by
ballasting of aggregates (Abrantes et al., 2002;Armstrong et al., 2002;Armstrong et al., 2009;Bárcena et al., 2004;Berger et
al., 1989;Engel et al., 2009;Iversen and Ploug, 2010;Milliman, 1993;Ridgwell and Zeebe, 2005;Ziveri et al., 2007).
Consequently, there is a wide interest in the response of coccolithophorid calcification to changes in ocean productivity and
carbon system. To move beyond the responses of calcification in clonal cultures in short duration experiments, recent efforts
have focused on characterizing the variations in coccolithophorid calcification from measurements of the thickness of
coccoliths in populations from the water column and ocean sediments. Such measurements are effectively made using
images of coccoliths taken by polarized light microscopy, for which the interference color varies with the thickness of the
calcite in the coccolith (Beaufort, 2005). Recent advances using circular polarized light (Bollmann, 2014;Fuertes et al.,



2014) permit such calculations with a single image rather than composite at several orientations as was done in previous approaches (Beaufort et al., 2014).

Until recently, coccolith thickness measurements have been limited to thicknesses less than 1.55 μm, for which birefringence remains in grayscale tones. Beaufort et al. (2014) suggested that measurements might be extended to 4.5 μm thickness by

employing variations in the colored birefringence range using a set of equations in HSL color system.

While there is a clear theoretical dependence of interference color on calcite thickness, the particular grayscale or color value obtained for a given calcite particle depends also on the microscope configuration, light intensity and spectra of the light employed at the time of measurement. Even for a given microscope system this relationship changes over time due to bulb aging and microscope settings, so a daily calibration is required. However, this calibration is challenging due to the absence

of reference materials, for which the observed interference color on a given day may be related to a known thickness. The situation becomes even more complex for coccoliths thicker than 1.55 μm, as there is no longer a single bivariate theoretical curve between grayscale and thickness; rather representation of thickness depends on colors registered as three parameters (eg RGB, or HSL, or HSV). Beaufort et al. (2014) proposed a set of equations to relate HSL color values to thickness in the range of 1.55 to 4.5 μm, but these particular equations may be specific to particular light settings and manufacturer,

configurations of a particular microscope and thus should not be universally applied.

Here we discuss a calibration option which is relevant for calibration in the grayscale and color region, and readily allows each user to establish relevant equations for his/her own microscope system, which may be modified to account for effects such as bulb aging and light spectra. We describe the preparation and use of a calcite wedge for which thickness can be independently constrained, and which can then be used as a daily calibration material for both grayscale and color scale. In

this work, following previous studies (Bollmann, 2014;Fuertes et al., 2014), we employ circular polarized light (Bass, 2009;Frohlich, 1986;Hecht, 2002;Higgins, 2010) so that our interference colors are independent of the calcite c-axis orientation and coccoliths are viewed without extinction patterns. With this calcite wedge, user can calibrate microscope-specific equations relating color to thickness.

## 2 Background on use of the polarized circular light and details of previous calibrations

Anisotropic minerals, such as the calcite comprising coccolithophores, modulate the vibration direction of light passing through them. When light is passed through an anisotropic (birefringent) mineral, the light is split into two rays of different velocity. The retardation of the slow ray relative to the fast ray increases with increasing thickness of the mineral. For a given mineral, the equation relating thickness (t) and retardation (r) is given by the Eq. (1) (Delly, 2003):

$$t = \frac{r}{b \ x \ 1000},$$    (1)

Where the birefringence (b) is a property of the mineral lattice (0.172 for calcite). Optically, this retardation is manifest by the appearance of interference colors when the light rays emerging from the mineral are recombined by passing through a second polarizing filter. For a given mineral orientation, progressive degrees of retardation produce a sequence of



interference colors first defined in the Michel-Levy chart, recently revised by Sørensen (2013). In the case of calcite, these ranges from black through gray to white at a thickness of 1.55 microns; then to yellow, red, and blue as thicknesses progressively increases up to 4 microns.

Several approaches have been used previously to ascertain coccolith thickness from measured grayscale values of coccoliths

in the thickness range from 0 to 1.55 μm. One approach has been application of a theoretical sigmoidal relationship between grayscale and thickness (O'Dea et al., 2014). Another common approach has been to assume a cylindrical shape of rods of common nannofossil genera *Rhabdosphaera clavigera*, so that rhabdolith diameter is taken to be the known thickness and the grayscale value corresponding to this thickness can be used for an empirical calibration or for pinning tie-points to a theoretical sigmoidal calibration (Beaufort and Dollfus, 2004;Beaufort et al., 2014;Fuertes et al., 2014). Alternatively,

Bollmann (2014) employed polymer sheets which produce a specified degree of retardation. The grayscale value corresponding to this degree of retardation can be understood to be the grayscale value that would be produced by a thickness of calcite which would produce a comparable retardation (estimated using Eq. (1)).

Our method with a calcite wedge permits calibration continuously over the thickness range from 0 to 4 microns, contributing to reproducible and accurate thickness measurements of a wide range of genera of modern and ancient coccoliths.

**3 Preparation and validation of calcite wedge**

**3.1 Manufacture of calcite wedge thin section**

Using a natural calcite crystal of Iceland spar, we have produced several calcite wedges. A plane free of fissures and defects was selected and a small rectangular prism was cut using a water cooled diamond saw. This fragment was abraded with carborundum-water slurry and polished and glued with epoxy resin to a glass slide, using pressure to minimize bubble

formation between the calcite and glass and heat to accelerate epoxy curing. The calcite was then cut to 1 mm thickness and further abraded to attain a 100 μm thickness. A wedge is attained on one border of the calcite through a final manual abrasion step, by applying a greater pressure on one edge of the slide to accentuate the removal rate of material there. During this final abrasion, the edges of the slide are evaluated using a microscope to verify when the borders enter into the first order interference colors corresponding to the thickness range from 0-4 microns (0 – 550 nm retardation).

Because the calcite fractures easily at thicknesses less than 10 μm, only a small portion of the slide – one or more zones on the edge of the calcite – forms a wedge in the relevant thickness range. The wedge which features the least number of imperfections in the calcite (fractures, bubbles or polishing grit, abrupt thickness changes due to breaks on cleavage planes) is adopted for calibration. We produced multiple slides of Iceland spar prepared in an identical way, and identified two as providing an acceptable area with a wedge in the desired thickness range in one part of the slide (Fig. 1). We protect the

wedge with a coverslip adhered with Canada balsam, in the same way as the coccolith slides are prepared, avoiding differences in the light absorption among them. Despite optimal precaution to minimize particles or bubbles, nonetheless some of these are present and generate halos which are visible in high resolution images. For work in the grayscale range, we



use a wedge produced at the University of Oviedo (OVD-W1). For work in the color zone, we employ a wedge produced at ETH Zurich (ETH-W2) which has fewer defects in the wedge in the transition to colors.

### 3.2 Microscope method

Digital images were obtained on two similarly configured microscope systems. All grayscale work was conducted at the
University of Salamanca using Nikon Eclipse LV100 POL microscope with circularly polarized light, equipped with a Nikon Plan Fluor 100X/1.30 oil OFN25 DIC H/N2 objective, a universal condenser with numerical aperture of 0.9 and a Nikon digital camera DS-Fi1. The camera resolution is 1920x1280 giving a pixel resolution of 0.035 μm. For color work, images were obtained at ETH Zurich using a Zeiss Axioscope HAL100 POL microscope with circularly polarized light, equipped with a Zeiss Pan-APOCHROMAT 100x/1.4 Oil objective, a universal condenser with numerical aperture of 0.9 and a Zeiss
Axiocam 506 Color. The camera resolution was set to 2560x1920, resulting in a pixel resolution of 0.0454 microns. In all of these microscopes, for circular polarization, two λ/4 retardation plates have been placed in the microscope, one is located between the lower linear polarizer and the condenser and the other one above the upper linear polarizer (both at an angle of 45° relative to it) to transform the linearly polarized light into circularly polarized light (Fuertes et al., 2014). The microscope is turned on for 30-60 minutes to warm up and stabilize the light conditions before any images are captured. On
the Nikon, images were collected using the software specific to the camera 'Nis-Elements BR'. On the Zeiss microscopes, images were collected using 'Zeiss Zen 2.3 (blue edition)' imaging software.

### 3.3 Independent determination of thickness of the calcite wedge using a Tilting compensator

To determine the thickness more continuously along the wedge, we have used a Tilting Compensator B 0-5 Lambda, 6x20 (D) from Zeiss (Fig. 2A). This compensator is used under plane polarized light, and consists of piece of uniaxial birefringent
material (magnesium fluoride) cut normal to the optical axis that can be tilted about an axis parallel to the sample (in our case, the calcite wedge). Progressive rotation of the compensator about the horizontal axis generates a decrease in the retardation of the rays emerging from the observed calcite wedge. The tilting angle is read accurately from a calibrated micrometer drum and the corresponding decrease in retardation can be read from the corresponding tables supplied with the compensator. For the measurement of our calcite wedge, at maximum light intensity position, we have rotated the
compensator in increments corresponding to 0.25 μm, capturing a digital image at each increment to identify the distance along the wedge experiencing full compensation (negligible interference = black) (Fig. 2B). At the lowest rotation angle, the small decrease in retardation decreases interference colors only slightly, so the zone of full compensation (negligible interference = black) appears on calcite near the edge of the slide (Fig. 2C). With increasing rotation angles, the decrease in retardation produces a stronger decrease in interference colors, so the zone of negligible interference (black) moves
progressively up thicker parts of the calcite wedge towards the interior of the slide (Figs. 2C-E). Over a series of 16 incremental advances in angle, we quantify the location of the compensation point (Fig. 2E) to derive the relationship between thickness and distance from the edge of the wedge.



### 3.4 Validation of the thickness using polymer retarders

To validate the thickness profile obtained from the rotating compensator, we have used high precision plastic polymer retarders from Meadowlark Optics with a wavelength of 550 nm and retardations of 136.6 and 274.8 nm, which give retardations equivalent to calcite with thickness of 0.79 and 1.59, respectively. Each was mounted in Canada balsam in

separate glass slides. On half of each retarder (Fig. 3A) we decanted a suspension of modern coccoliths using the technique of Flores and Sierro (1997). This provides a point to initially focus on each slide (Fig. 3B). Then, we move to an area of the retarder without the coccoliths (Fig. 3C) and increase the light intensity until we reach saturation on the polymer (Fig. 3D). Without changing conditions, we place the calcite wedge under the microscope, make a precise focus, and in the digitally captured image allow the software to map light saturation (defined as sum of RGB as 256/256). The lowest point on the

wedge at which saturation is attained corresponds to the thickness of the calcite wedge with a retardation identical to that of the plastic polymer (Fig. 3E). We repeat the procedure with the other polymer (Fig. 3F). Using the known birefringence of calcite, and the known retardation of each polymer, we can use Eq. (1) to estimate the thickness of the calcite at each of these two points.

### 3.5 Thickness profile of predefined wedge

For subsequent calibration of the relationship between interference color and thickness, we have defined a working profile across each wedge which is referenced in our image analysis routine (ImageJ-Fiji). Along this profile in OVD-W1, the wedge reaches a 4 µm thickness at a distance of 42 µm from the start of the profile in the OVD-W1; in ETH-W2 the 4 µm thickness is reached at a distance of 28 µm from the start of the profile (Fig. 4). The validation points from the retarder polymers fall along the line of thickness calculated from the tilting compensator. All subsequent calibration efforts describe

the relationship between interference colors and thickness along these fixed profiles.

### 4 Application to thickness measurements in the grayscale range

Over the thickness range from 0 to 1.55 the calcite wedge exhibits the expected increase in grayscale, with the expected sigmoidal tail at the lowest thickness values (Fig. 5A). The exact grayscale value for a given thickness is dependent on the light setting and exposure times, but for a given setting is highly reproducible day to day (Fig. 5A).

For measurements of coccolith thickness, optimal sensitivity can be obtained when the light setting is adjusted for the range of thicknesses in the samples of interest; for example, higher light and exposure times for samples with very thin coccoliths. This is particularly useful with 8-bit cameras which have a lesser range of definition than 14 or 16 bit cameras. By matching the light and exposure settings used for a set of samples with those used to capture an image of the calcite wedge, we develop a robust calibration between grayscale and thickness under any illumination parameters. For routine calibrations of

grayscale to thickness, we select 10 points along the wedge profile, with known thickness established as in Fig. 4.



### 4.1 Example application of the calibration system to cultured coccoliths

We have used the calcite wedge to calibrate coccolith thickness measurements in coccoliths produced by 8 strains of coccoliths in controlled laboratory culture conditions (as described in (Bolton et al., 2016)). Image processing was carried out by C-Calcita MATLAB routine described by Fuertes et al. (2014). The C-Calcita routine permits either a linear or

sigmoidal regression between grayscale and thickness. The linear option, with a zero intercept, has been employed in previous studies (Beaufort, 2005) and was employed in our original processing of the culture sample images (Bolton et al., 2016). Recent updating of the Michel-Levy curve (Sørensen, 2013) suggests that in the first order interference range the grayscale thickness relationship is better represented with a sigmoidal curve, an approach adopted by recent coccolith thickness studies (Beaufort et al., 2014;O'Dea et al., 2014). An updated version of C-Calcita routine now permits sigmoidal

calibration using an arcsine function to fit the calibration points made from the calcite wedge.

For the same image sets and calibration points from the calcite wedge, we have compared the thickness estimated from the linear vs sigmoidal calibration slopes for three of the culture samples. Given that a high proportion of the coccolith has a very low thickness and falls within the sigmoidal part of the calibration relationship, the sigmoidal calibration yields average thickness estimates for coccoliths which are nearly twice those attained with a linear calibration (Fig. 5B,C). For thicker

coccoliths, where a greater percentage of the individual coccolith falls within the linear portion of the sigmoidal curve, the difference is not expected to be as significant.

### 4.2 Grayscale vs thickness relationship in the calcite wedge and *Rhabdosphaera clavigera*

We compare the relationship between grayscale and thickness in the calcite wedge with that of *R. clavigera* nannoliths, applying the conventional assumption that they rods are cylindrical and therefore the diameter is equivalent to the thickness.

A total of 15 rhabdoliths were measured under identical light and exposure conditions, and for each rhabdolith we made 10 measurements of width/thickness and its corresponding gray level at different points. We observe that for the same estimated thickness, the gray level values obtained show great variations among the chosen rhabdoliths. The regression slopes between estimated thickness and grayscale for each rhabdolith show a large range (Fig. 6A), although if each is forced to pass through the coordinate origin (a premise for applying a calibration), the slopes converge slightly (Fig. 6B; Table I). For the

same microscope conditions, the individual rhabdolith thickness-grayscale coordinates fall significantly above and below the relationship defined by the calcite wedge (Fig. 6B). The relative standard deviation of 40% to 50% on the regression slopes from different rhabdoliths is very large compared to the range of thicknesses among different coccolith populations from culture. Measurement of the same rhabdolith for calibration for every session would ensure internally reproducible thickness measurements. However, the absolute thickness determinations may be biased by the geometry of the particular rhabdolith

specimen used for calibration. On the other hand, use of different rhabdolith specimens for calibration for different measurement sessions could lead to significant variations in thickness results with both poor reproducibility and poor



accuracy. If our population is representative, statistically, measurement of at least 18 individual rhabdolith specimens in each session would be required to reduce the standard error of the mean (calibration slope) to 10% (Table I).

## 5. Application to thickness measurements in the color range

### 5.1 Use of calcite wedge to develop calibration equations in color range

Whereas small placoliths of the modern and late Pleistocene oceans possess coccoliths thinner than 1.55 µm, in the grayscale range, the large modern and ancient placoliths possess thicker coccoliths which enter into interference colors. For this reason, it is fundamental to develop approaches for quantifying thickness beyond the grayscale range. Due to the high density of defects in the color region of OVD-W1, we employ ETH-W2 for calibration of the relationship between thickness and interference colors. We obtain a color image of the calcite wedge, ensuring that light intensity and camera exposure time

are adjusted to give light saturation at the previously calculated point where the wedge reaches 1.55 µm of thickness. For the development of calibration equations, we sample 1 of every 10 pixels along the calibration profile, leaving ample independent data for subsequent validation.

    Digital images from color cameras encode color variations using an additive synthesis of three primary light colors red, green and blue (RGB). We employ *"Color Transformer"* plugin from ImageJ-Fiji to convert RGB values along the

calibration profile to two alternative color models: HSV (Hue, Saturation, Value) and HSL (Hue, Saturation, Lightness; Fig. 7). To identify useful regressions between the components of these color models and thickness, we calculate the correlation between the wedge thickness and each of these individual color components as well as combinations of them, over the entire wedge and over discrete sectors (0.5 microns wide) of the wedge (Table II). We find slightly higher correlations in the HSV color model than the HSL, and subsequently use HSV. Although for any given sector of the wedge, there is a color

component highly correlated with thickness, no single color component maintains a constant and high sign of correlation with thickness over the entire wedge profile. Therefore, the development of a model to relate thickness to the color components will require 1) alternating between a series of equations, and 2) defining criteria that uniquely specify, based on the observed color components, which equation is to be used. The definition of these criteria or thresholds is the most challenging part of successfully applying equations to calculate thickness.

From the variation of color components in our calcite wedge, we find that between 0 and 4 µm, four distinct cases, each with their own equation for thickness, can be defined (Fig. 7A; Table III). The first case is defined as pixels meeting either V value below 130, or H values between 110 and 160 while S<80 or V<170. It encompasses the thickness range between 0 and 1.4 µm, over which the V index increases with increasing thickness (correlation with thickness of 0.99). The second case is defined by H values between 120 and 19 while V>120. It encompasses the thickness range between 1.4 and 2.5 µm, over

which the S-V combination provides optimal correlation with thickness (0.96). This component, and all others, show a lower sensitivity to thickness between 1.5 and 1.7 µm; the S-V component maintains best sensitivity to thickness in the range 2.2 to 2.4 µm. A third case is defined by H<19 and V>150. This encompasses the thickness range from 2.5 to 3 µm, over which the




V index shows a high negative correlation with thickness (-0.98). A final interval is defined for all pixels not meeting the criteria of the other three categories. This interval includes the thickness range 3-4 µm, over which the H index shows a strong and negative linear correlation with thickness (-0.97). While the correlation table shows linear regressions, in detail the best fits are in some cases polynomials (Fig. 7B). Equations derived from these polynomial fits are ultimately used for

the calculation of thickness from the various color indexes, with attention paid to employing at least 5 significant figures in the coefficients of the equations to prevent rounding errors.

We validate this calibration by comparing the observed and calculated thickness of pixels along the calibration line of the ETH-W2 calcite wedge, using pixels which were not employed to generate the regression equations (Fig. 7C). Results show a continuously robust estimation of thickness throughout the entire range from 0 to 4 microns. Besides, using open source

software ImageJ-Fiji we analyzed images of both wedge profile regions and applied the conditions and equations to validate the thickness with the birefringence color (Fig. 7D).

These particular equations may not be exportable to other different microscopy and camera systems, or other camera or light configurations on a same microscope. For example, a previous publication (Beaufort et al., 2014) proposes a set of three equations employing the HSL index for calculation of thickness between 0 and 4.5 µm. Using the validation data set from

our profile along the calcite wedge, we calculate thickness using the exact equations and thresholds provided by Beaufort et al. (2014). The thickness calculated from those equations deviates significantly from the actual calcite wedge thickness along several portions of the calcite wedge, in particular in the ranges 1.4-1.7 µm and 3-4 µm (Fig. 8). This result shows that both the thresholds and equations relating color to thickness must be calibrated on each microscope system and settings. We suggest that several factors may cause variation in the color components for a given thickness. First, the spectrum of the

microscope light source will vary the intensity at different color wavelengths and this may vary both among microscopes and over time due to bulb aging. Secondly, the use of filters, as well as objective characteristics, diaphragm aperture, light intensity, and light absorption by slides within the microscope system may affect the color components for a given thickness. Finally, within the digital camera, the quantum efficiency for a given wavelength may be different for different camera detectors. For example, the Zeiss camera employed here reports a very high efficiency in the green region compared to red

or blue, whereas other cameras report a more similar efficiency among the wavelengths. Adjustments in the white balance in the image analysis system can in theory make the color components more similar among different cameras, but in practice we find that there is still variation that requires adjustment to the calibration. Even small variations in the calibration equations become very important in the estimation of thickness in coccoliths, as described in section 5.2.

Since the relationship between interference colors and thickness needs to be established for each particular microscope-

camera configuration and validated routinely, we propose that a calcite wedge may serve as one such optimal calibration material. For a new microscope system, we advise capture of digital color image of a calcite wedge for which the profile of thickness has been independently determined from the tilting compensator. A first step would be to graphically represent thickness and color components to determine zones of useful relationships between them. A second step would be to identify the unique thresholds which could be used to distinguish portions of the thickness range requiring different equations. In





HSV, one would expect a similar basic sequence of changes in color parameters with increasing thickness, however, the exact values of the threshold parameters can change appreciably with small changes in microscope or camera settings. The most significant challenge of calibration lies in identifying the thresholds, because errors in threshold definition result in application of the inappropriate equation for calculating thickness for that pixel and thus lead to large errors in estimates of thickness in coccoliths. The saturation of light at a thickness of 1.55 μm is also crucial since it allows setting the boundary between the gray domain (black to white) and the color range and can be verified either with synthetic polymers or prior mapping of the zone of such thickness on the calcite wedge. In our experience, it is important to define saturation as the first appearance of grayscale values of 256 in the digital image (taken with an 8-bit camera), rather than relying on the mapping of saturation by the image analysis system. Oversaturated images can produce blank areas and peaks in H values between the thresholds for our case 1 and case 3, which leads to an overlap in criteria for thicknesses in the 1.3 to 1.4 μm and 3.2 to 3.3 μm thickness range, and misapplication of the case 4 equation to pixels in the 1.3 to 1.4 μm thickness range.

**5.2 Application to ancient *Reticulofenestra* specimens and modern *Helicosphaera carteri* coccoliths**

In order to test calibration equations and thresholds, we have implemented a macro in ImageJ-Fiji to automate the calculation of thickness of coccoliths as well as geometric parameters. This tool reads pixel values from a selected specimen and applies the appropriate regression equations to each pixel, creating a thickness map of the coccolith.

For this test, we have observed slides, settled by Flores and Sierro (1997) method, from IODP Expedition 342, where preservation of coccolithophores is moderate to good (Norris et al., 2014) and thickness in some parts of the Oligocene coccoliths exceeds 1.55 μm. We have complemented this check with Zygodiscales *Helicosphaera carteri* taxa of Late Quaternary age from NIOP 905.

Several sample thickness maps and profiles for *Reticulofenestra spp* and *H. carteri* individuals are shown in Fig. 9A-E. Tridimensional schemes represent total thickness accumulated across the coccolith crystal units for which the optical axis is not parallel to the light ray, that is with radially-oriented axes. Essentially all of the *Reticulofenestra* calcite is radially oriented and thus effectively quantified. In contrast, the *H. carteri* coccoliths feature both radially oriented and vertically oriented calcite, of which the thickness of the radially oriented calcite is accurately calibrated by the calcite wedge equations.

Although the surface plot is not smooth, the cross sections profiles show a pattern that can be coherently related with the main thickness configuration plots (Fig. 9F), built from the models of Young et al. (2003). From such maps, the mean coccolith thickness may be calculated and permit estimates of evolution of calcification of large coccolith species over time and in different locations.

**6 Useful online resources**

http://fiji.sc/Fiji
http://www.microscopy-uk.org.uk/mag/artmay14/jp-retardation.pdf
http://www.microscopyu.com/
http://www.mikrotax.org/Nannotax3/

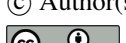


http://www.modernmicroscopy.com/
http://zeiss-campus.magnet.fsu.edu/

**7 Conclusions**

The ability of the scientific community to compare optically determined estimates of coccolith thickness among different
laboratories will depend on the reproducibility of the methods for calibration of relationships between thickness and light
intensity. A detailed survey of the behaviour of one commonly used calibration material, *R. clavigera*, showed that the
calibration is highly dependent on the particular calibration specimen used. In addition, calibrations in the 1.55 to 4 μm
thickness range require a calibration material to accurately account for differences in color spectra under different
measurement conditions. We propose a new calibration method based on the development and use of a wedge of calcite to
relate light intensity recorded with the coccolith thickness. The thickness of the calcite wedge can be independently
calibrated using a tilting compensator, and validated using polymers of known retardation. The calcite wedge calibration has
been applied to measurement of thickness of cultured coccoliths from several strains of coccolithophorids, and allowed to
examine specimens to quantify the >1.55 μm thickness whose birefringence spans beyond the greyscale into the color range.
In future, the use of image analysis software could permit the development of macros to further automate the calibration
process.

**Acknowledgements**

This research was funded by the European Research Council under Grant ERC-STG-240222 PACE to H.M.S. We thank
Andrés Cuesta and Ángel R. Rey for advice on manufacture of the calcite wedge and use of the Berek and tilting
compensators, and Ángel M. Nistal for assistance in macro development in ImageJ-Fiji.

**Author contributions**

This study was conceived by H.M.S. Measurements were conducted by S.G.L. and J.G., with guidance from J.A.F. and
M.A.F. Calculations and analysis was conducted by S.G.L., H.M.S., and J.G.; S.G.L, J.G., and H.M.S. wrote the paper, with
input from other authors.

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

Table I. Linear and sigmoidal calibrations for a 15 rhabdolith stems.

| Rhabdolith | Sigmoidal calibration | Linear calibration |
|---|---|---|
| R1 | 0.0049 | 0.0106 |
| R2 | 0.0048 | 0.0140 |
| R3 | 0.0113 | 0.0252 |
| R4 | 0.0091 | 0.0242 |
| R5 | 0.0087 | 0.0204 |
| R6 | 0.0042 | 0.0078 |
| R7 | 0.0074 | 0.0188 |
| R8 | 0.0030 | 0.0070 |
| R9 | 0.0041 | 0.0065 |
| R10 | 0.0024 | 0.0054 |
| R11 | 0.0063 | 0.0156 |
| R12 | 0.0062 | 0.0101 |
| R13 | 0.0058 | 0.0124 |
| R14 | 0.0066 | 0.0112 |
| R15 | 0.0039 | 0.0065 |
| Max | 0.0113 | 0.0252 |
| Min | 0.0024 | 0.0054 |
| Mean | 0.0059 | 0.0131 |
| Standard deviation | 0.0024 | 0.0065 |
| Uncertainty (%) | 41.43 | 49.93 |



Table II. Correlation between the calcite wedge (ETH-W2) thickness and each of these individual color components, as well as combinations of them, over discrete intervals (0.5 microns).

| Thickness Range (µm) | R | G | B | RGB | H (f256) | S (f256) | V (f256) | H/S | H/V | S/V | H+S | H+V | S+V | H-S | H-V | S-V | H+S+V | H (f256) | S (f256) | L (f256) | H+S | H+L | S+L | H-S | H-L | S-L |
|---|---|---|---|---|---|---|---|---|---|---|---|---|---|---|---|---|---|---|---|---|---|---|---|---|---|---|
| 0 – 0.5 | 0.998 | 1.000 | 0.999 | 1.000 | -0.900 | 0.897 | 0.999 | -0.852 | -0.962 | 0.081 | 0.833 | 0.997 | 0.989 | -0.919 | -0.999 | -0.978 | 0.988 | -0.900 | 0.897 | 0.999 | 0.706 | 0.996 | 0.996 | -0.926 | -0.999 | -0.990 |
| 0.5 – 1 | 0.995 | 0.998 | 0.996 | 0.999 | -0.205 | -0.586 | 0.996 | 0.583 | -0.967 | -0.900 | -0.795 | 0.995 | 0.959 | 0.395 | -0.990 | -0.992 | 0.975 | -0.205 | 0.969 | 0.998 | 0.976 | 0.991 | 0.995 | -0.938 | -0.997 | -0.964 |
| 1 – 1.5 | 0.986 | 0.959 | 0.859 | 0.960 | -0.855 | -0.728 | 0.972 | -0.020 | -0.905 | -0.772 | -0.900 | -0.718 | 0.935 | -0.794 | -0.918 | -0.940 | -0.802 | -0.855 | 0.732 | 0.960 | 0.234 | -0.680 | 0.830 | -0.810 | -0.930 | 0.574 |
| 1.5 – 2 | 0.000 | -0.734 | -0.975 | -0.951 | -0.505 | 0.975 | 0.000 | -0.962 | -0.505 | 0.975 | 0.987 | -0.505 | 0.975 | -0.942 | -0.505 | 0.975 | 0.987 | -0.505 | 0.000 | -0.975 | -0.505 | -0.903 | -0.975 | -0.505 | 0.950 | 0.975 |
| 2 – 2.5 | -0.800 | -0.996 | -0.996 | -0.995 | -0.972 | 0.988 | -0.800 | -0.945 | -0.946 | 0.997 | 0.989 | -0.955 | 0.924 | -0.987 | 0.187 | 0.997 | 0.901 | -0.972 | -0.804 | -0.993 | -0.875 | -0.998 | -0.912 | 0.690 | 0.971 | -0.228 |
| 2.5 – 3 | -0.978 | -0.997 | 0.191 | -0.994 | -0.954 | -0.812 | -0.978 | -0.953 | -0.930 | -0.315 | -0.868 | -0.974 | -0.924 | 0.632 | 0.977 | -0.931 | -0.931 | -0.954 | -0.977 | -0.980 | -0.975 | -0.992 | -0.985 | 0.979 | 0.394 | -0.960 |
| 3 – 3.5 | -0.987 | 0.792 | 0.986 | 0.715 | -0.983 | -0.903 | -0.360 | 0.340 | -0.783 | -0.918 | -0.991 | -0.989 | -0.758 | -0.836 | -0.866 | -0.822 | -0.974 | -0.983 | -0.749 | 0.167 | -0.992 | -0.996 | -0.497 | -0.810 | -0.929 | -0.954 |
| 3.5 – 4 | -0.918 | 0.979 | 0.975 | 0.751 | -0.970 | 0.945 | 0.975 | -0.917 | -0.974 | 0.914 | 0.908 | -0.482 | 0.964 | -0.957 | -0.981 | 0.892 | 0.952 | -0.970 | 0.974 | -0.600 | 0.958 | -0.933 | 0.978 | -0.976 | -0.943 | 0.960 |

Table III. Definition of four unique or combination color components and the relevant equations for the thickness calculation for our microscope conditions. The calibration was established from the calcite wedge ETH-W2.

| | Thickness Range | Thresholds | Thickness proportional to | Equation |
|---|---|---|---|---|
| Case 1 | 0 – 1.4 µm | (110<H<160 & (S<80 or V<170)) or V<130 | Rising V | $T = 8.806\text{E-}10x^4 - 4.086\text{E-}07x^3 + 7.336\text{E-}05x^2 - 1.382\text{E-}03x + 7.080\text{E-}02$ |
| Case 2 | 1.4 – 2.5 µm | 19<H<120 & S<200 & V>120 | Rising S-V | $T = -1.171\text{E-}08x^4 - 7.856\text{E-}06x^3 - 1.961\text{E-}03x^2 - 2.102\text{E-}01x - 5.722\text{E+}00$ |
| Case 3 | 2.5 – 3 µm | H<19 & S>0 & V>150 | Decreasing V | $T = 2.318\text{E-}07x^4 - 1.930\text{E-}04x^3 + 6.000\text{E-}02x^2 - 8.260\text{E+}00x + 4.281\text{E+}02$ |
| Case 4 | 3 – 4 µm | ELSE | Decreasing H | $T = 2.164\text{E-}08x^4 - 1.881\text{E-}05x^3 + 6.034\text{E-}03x^2 - 8.527\text{E-}01x + 4.842\text{E+}01$ |



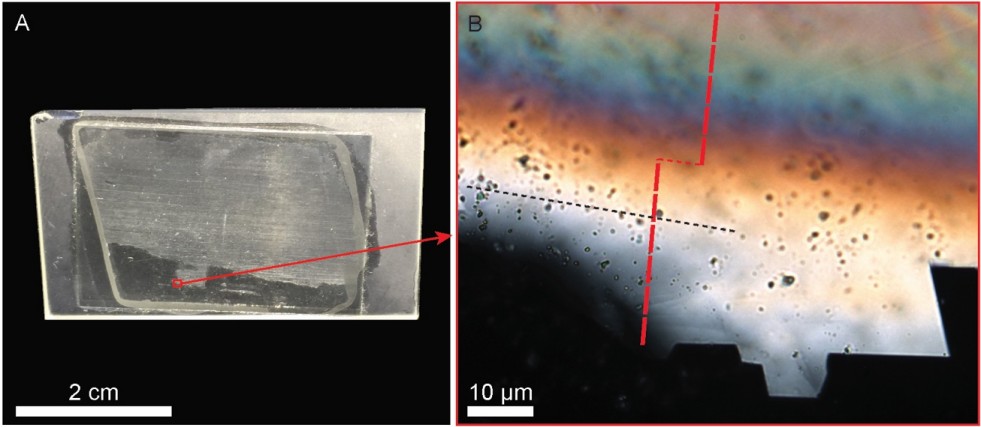

**Fig. 1.** OVD-W1 calcite wedge used for calibration. **A)** Photograph of the thin section of calcite protected under a cover slip. The red box indicates the region containing a wedge in the thickness range 0 to 4 microns. **B)** Digital image of the calcite wedge observed under circular polarized light at 1000x magnification (100x objective and 10x ocular). Red long-dashed lines indicate the profile along the wedge used for calibration. The dashed black line shows the approximate boundary between the grayscale interference colors and those entering into the color range.

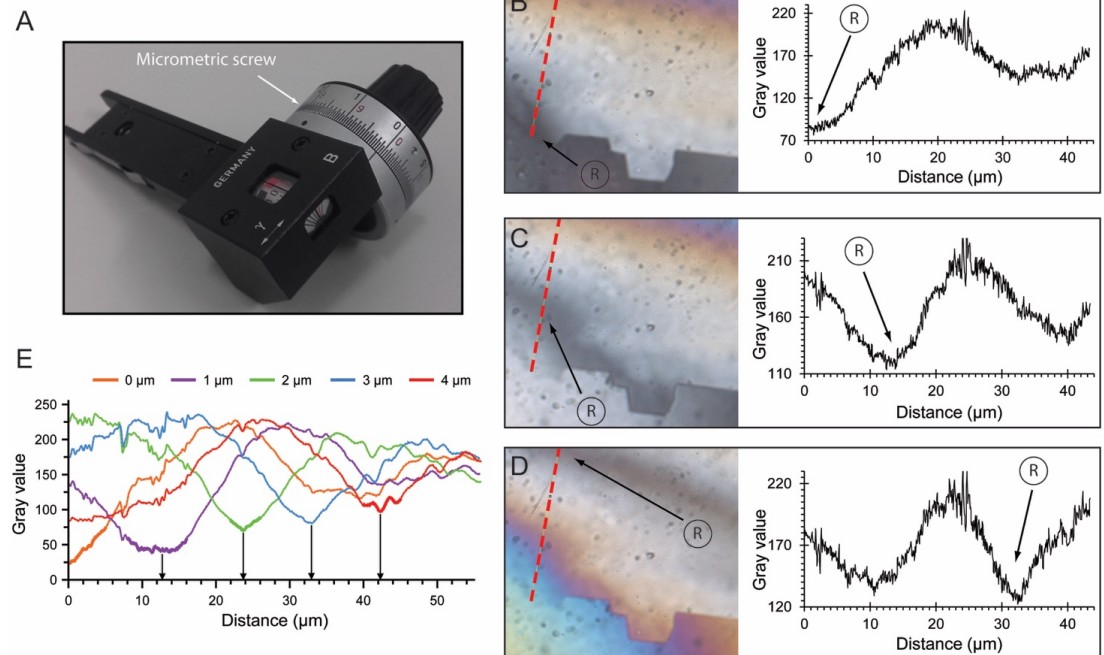





**Fig. 2.** Technique for quantifying wedge thickness. **A)** Tilting compensator B 0-5 Lambda. **B-D)** Digital images under linearly polarized light of the calcite wedge with successive rotations of the compensator. The minimum grayscale value in each image corresponds to the full compensation of interference colors at that point, and is designated by an "R" in each image. The blue tone produced by partial compensation of interference colors leads to second local minima in grayscale on either side of the full compensation. **E)** Full grayscale
5   profiles along the wedge with compensator rotation equivalent to four selected thicknesses of calcite. Arrows show the distance of maximum compensation for each tilting compensator rotation, and denote the distance on the OVD-W1 wedge corresponding to that thickness.

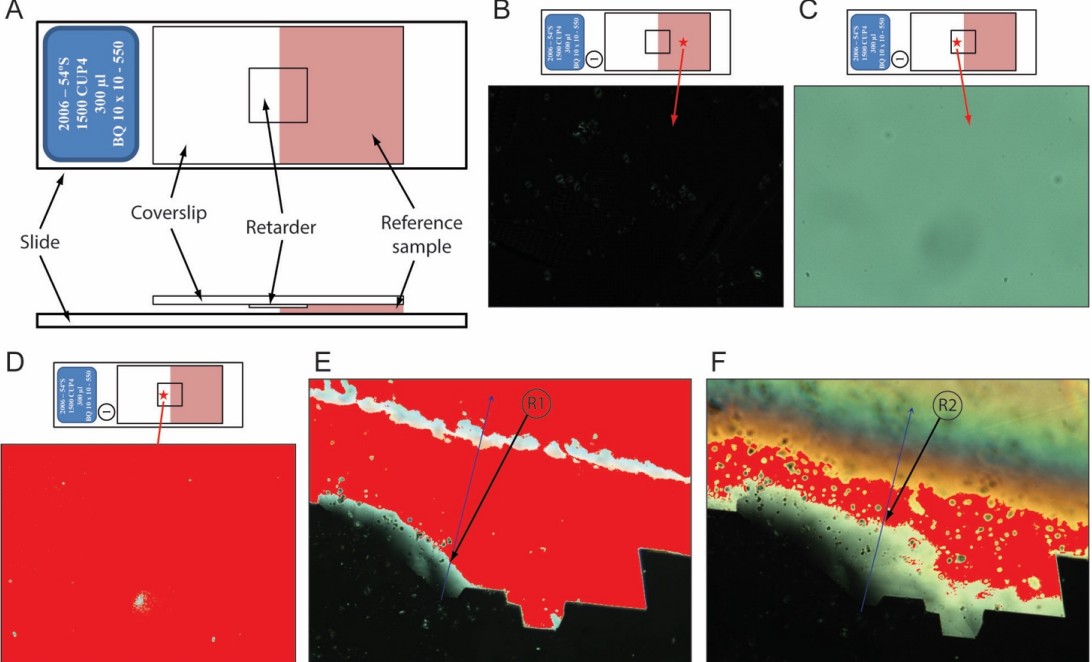

**Fig. 3.** Technique for validating wedge thickness at two tiepoints. **A)** Distribution of a retarder polymer and reference sample on a slide.
10   Only half of retarder is covered by the reference sample. **B)** Focus the coccoliths of the reference sample, **C)** move to the sample-free part on the retarder polymer, and **D)** increase the light intensity until polymer saturation (the image acquisition software indicates in red the saturated area). **E-F)** Wedge-calcite saturated at the same conditions as the retarders 1 and 2, respectively. The arrows (R1, R2) indicate the points where the thickness has been calculated.



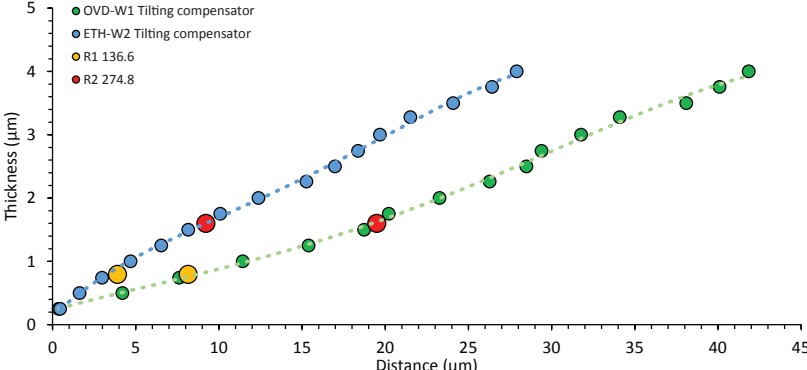

**Fig. 4.** Profile of the calcite wedges OVD-W1 and ETH-W2, as defined by the tilting compensator and the tiepoints established from the thickness-interference color relationship of the two polymer retarders (R1 and R2).

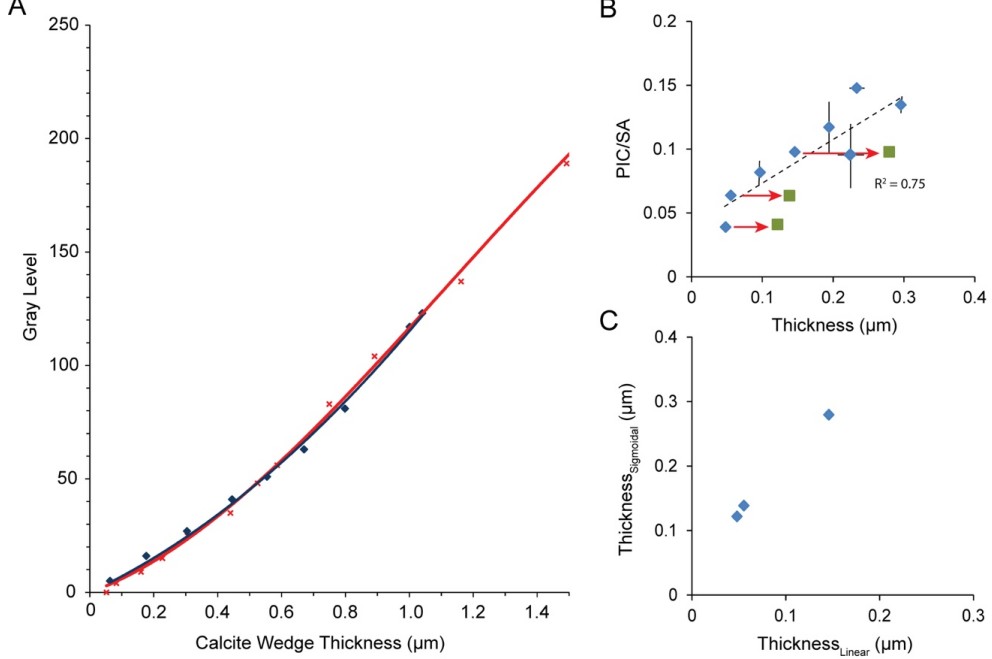

5  **Fig. 5. A)** Gray level vs. calcite wedge thickness on images taken with 8-bit camera. The wedge shows the expected sigmoidal shape at low thicknesses. The reproducibility of the calcite wedge, on images taken under very similar microscopic conditions on different moments, is very consistent. The blue line represent a calibration carried out on 10/29/2014, while the red line was made on 07/15/2015. **B)** Comparison of *Emiliania huxleyi* coccolith thickness estimates using linear and sigmoidal calibration forms. The graphic shows the relationship between cellular calcification (calcite per cell surface area) and coccolith thickness using linear calibration (blue symbols) and
10  sigmoidal calibration (green symbols, connected by red arrows to the equivalent sample measured with linear calibration). **C)** Comparison of linear and sigmoidal calibration for three populations of coccoliths from culture.



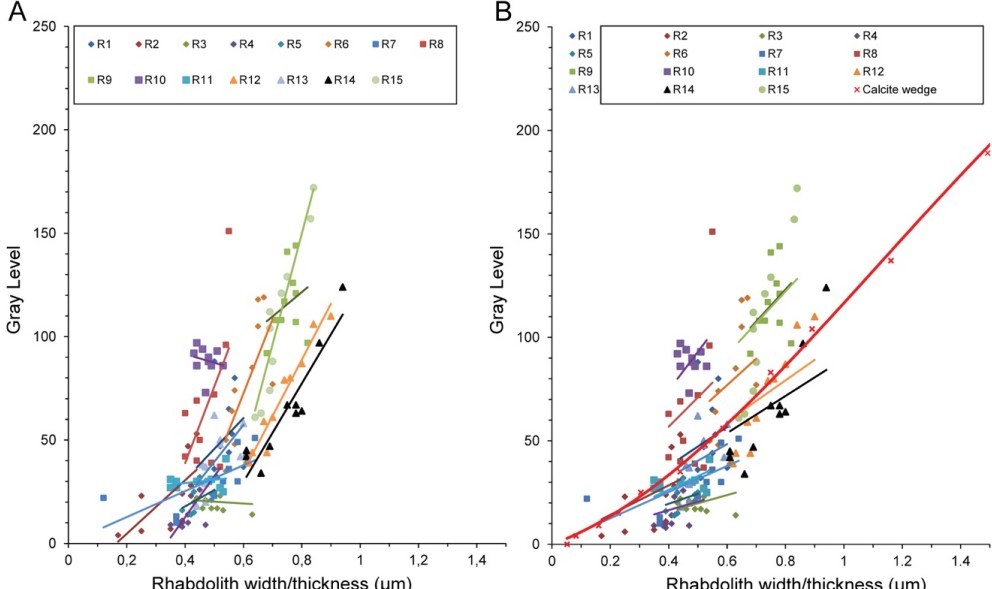

**Fig. 6.** Relationship between grayscale and thickness for 15 individual *Rhabdosphaera clavigera* rods (R1 to R15), and thickness for the calcite wedge (red line) measured under identical light conditions. The rhabdolith thickness is assumed to the diameter of the rod at the point of grayscale measurement. **A)** Natural trend of different rhabdoliths. **B)** Rhabdoliths forced to pass for the coordinate origin
5  (requirement for the calibration software).



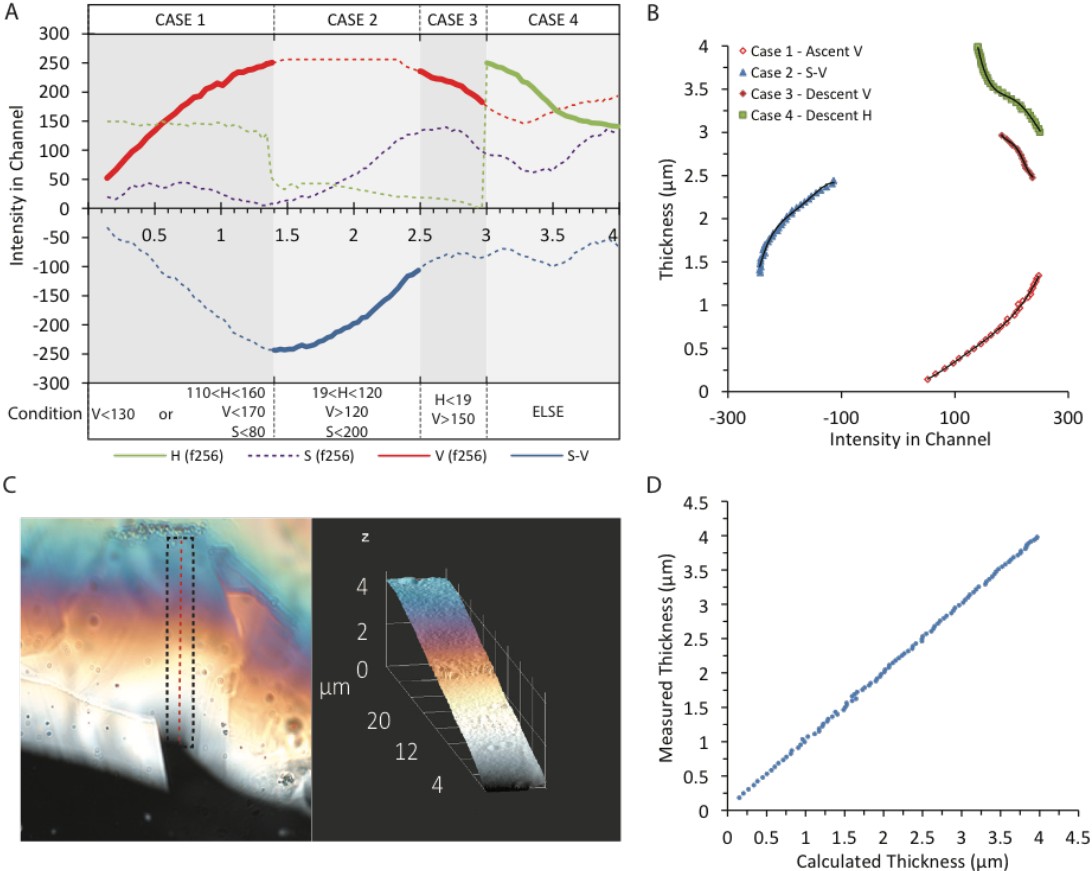

Fig. 7. **A)** Variation in components of color model HSV with increasing thickness along the calcite wedge profile. Solid lines indicate the range in which that color component is used to calculate thickness. **B)** Polynomial distribution for each interval. Table III gives the equations of each color component. **C)** Thickness tridimensional plots of calcite wedge area generated with image analyzer software using criteria and equations described on the text. **D)** Validation profile of the measured thickness of the calcite wedge ETH-W2 and the thickness calculated using the regression equations described in Table II. For the validation data, we used a different set of pixels than those used for the calibration equations





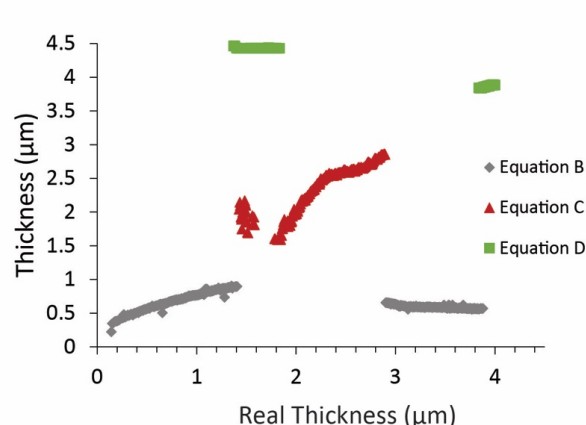

**Fig.8** Relationship between measured thickness (with tilting compensator) of the ETH-W2 calcite wedge and thickness calculated by application of the equations of Beaufort et al. (2014) for calculation of calcite thickness from HSL color model, for the validation dataset from the calcite wedge. The colors indicate the equation specified for use according to the criteria of Beaufort et al. (2014).



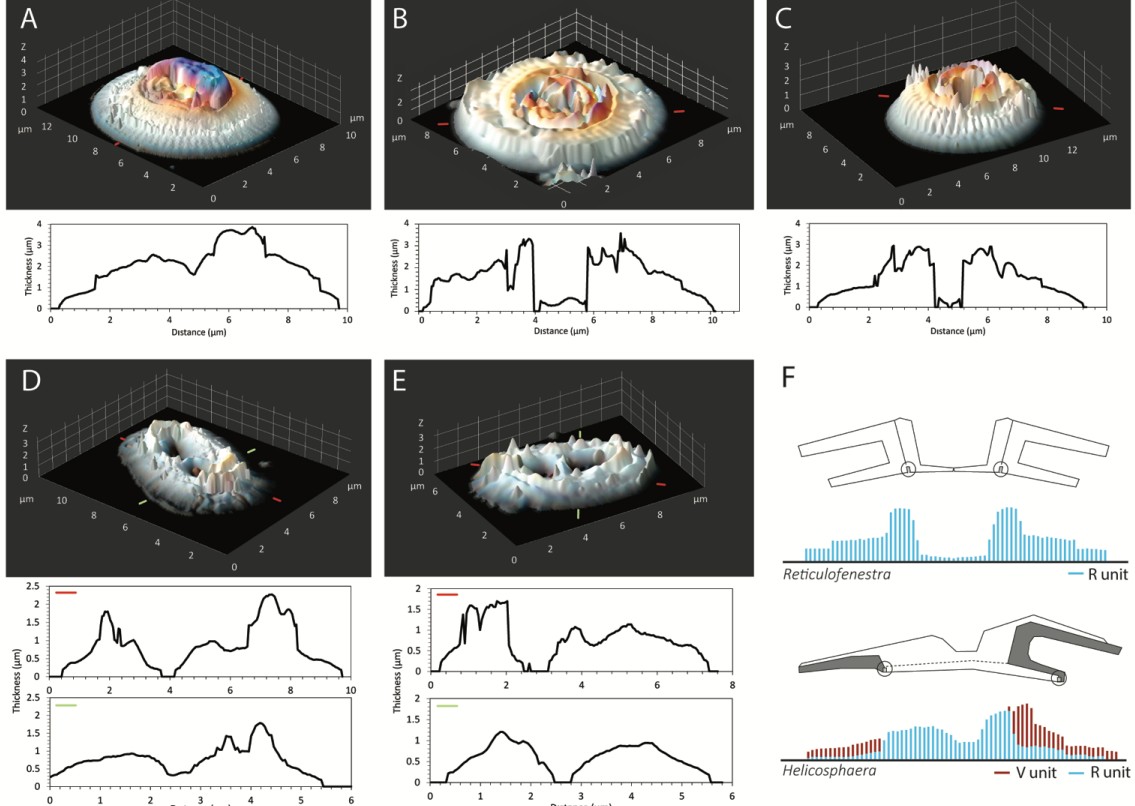

**Fig. 9**. Sample accumulated thickness schemes overlapped with color image and cross section profiles for *Reticulofenestra bisecta* **(A)**, *Cyclicargolithus floridanus* **(B-C)** from U1406A, and for *Helicosphaera carteri* **(D-E)** from 905. **F)** Thickness representation of two main coccolithophore groups used here from its cross-section shape schemes modified from Young et al. (2003).

