# Peer review of "An empirical method for absolute calibration of coccolith thickness"

_Biogeosciences, 2017_

## Short Comment (SC1) · 17 Aug 2017

Page 4, line 12: "one is located between the lower linear polarizer and the condenser and the other one above the upper linear polarizer" the other one should be below the upper linear polarizer.
* * *
[Figure]

---

## Short Comment (SC2) · 31 Aug 2017

You are right. As described in Fuertes et al., 2014 :

"A linear polarizer microscope can be transformed into a circular polarizer microscope by using two $\lambda/4$ retardation plates (Fig. 1a), placing one of them above the lower linear polarizer at an angle of 45° relative to it and the other one before the upper linear polarizer, also at 45°."

So, as you say, the other $\lambda/4$ plate should be below the upper linear polarizer, and not above it.

Thank you for your attention.

[Figure]

[Figure]

a

Linear polarizer

λ/4 plate

Objective

Slide

Condenser

λ/4 plate

Linear polarizer

Light source

**Fig. 1.** Fig. 1a

---

## Referee Comment (RC1) · Anonymous Referee #1 · 14 Sep 2017

This is a very well written paper that clearly sets out all the steps in obtaining an accurate calibration of thickness for coccoliths using polarization techniques in an imaging microscope. I found the paper easy to follow even for someone coming more from the optical physics side of the house. Every step is clearly described, all the sources of error are properly addressed and the procedures required to adapt other equipment and duplicate the fabrication of the wedge standard are clearly outlined ensuring that this paper will significant and of use to many other researchers in the field. I therefore recommend its publication. I only found one minor typo which is a testimony to the very good quality of the authors writing : Page 6 Line19 :"that they rods are cylindrical" should read "that they are cylindrical rods" The following comments are queries which come from my perspective as an optical physicist and should not in any way be

interpreted as a criticism of the paper. They may mostly be based on my ignorance of the customs and literature in the field. It seems to me that first glance that a relatively simple model for the polarization interference pattern as a function of illumination wavelength (even in the color range) of a given thickness of calcite could be derived as the retardation dispersion has been well measured.[ G. Ghosh. Dispersion-equation coefficients for the refractive index and birefringence of calcite and quartz crystals, Opt. Commun. 163, 95-102 (1999)]. In and of itself, this may not be precise enough for a calibration but it would be precious to have as a guide to the proper functional form of a calibration curve. As anyone considered using a well-chosen set of narrow band color filters in front of the illumination source of the microscope? The intensity of each wavelength for a given pixel could be recorded and the ratios calibrated against a blank slide response computed. These would give information on the differential retardation and therefore the thickness given the known indices of refraction of calcite. If a black and white camera where used this would be completely straightforward. If a color camera where used the camera color channel response matrix could be used for the same purpose. Finally, it seems to me that comparing the retardation map with an electron microscope size map could in principle be used to estimate the distribution of the c-axis crystal orientation in the lith structure. Has this been done by anyone?

---

## Referee Comment (RC2) · X. Jin (Referee) · 4 Oct 2017

This paper is concerning about a calibration between coccoliths thickness and their optical properties under circular polarized light microscope. And the usage of calcite wedges for the calibration provides a possibility and also a criteria to compare coccolith thickness and weight as estimated by their optical features when using light microscopes and cameras with different settings in other labs. The paper is well written, and authors provide a detailed technological processes for calcite wedges manufacture, and for measurements of the thickness of calcite wedges. So I see this study is important and recommend it can be published in Biogeosciences. Still, I have several questions that listed below: I have found inverse relations between grey level and width of Rhabdosphaera species (R9, R10) in figure 6A. Are these specimens poorly

preserved? These specimens may not be considered for calibration. It can be found that the linear K values of relation between grey level and width of Rhabdosphaera increase with the incensing of their width, when the width is <0.6 $\mu$m most Rhabdosphaera measurements are below the calcite wedge curve, and when >0.6$\mu$m most Rhabdosphaera are above the calcite wedge curve. So compared with the calcite wedge, all the Rhabdosphaera measurements make their calibration be more like a sigmoidal function. Is the background grey levels of the coccolith sample slides considered in the Rhabdosphaera calibration? And how about the grey level threshold for Rhabdosphaera coccoliths outline constrain? This is important for calibration, since for lighter/brighter part of Rhabdosphaera robs, their width could be overestimated due to dispersion. Technical corrections: Page 4 line 12: the other one "above"... below

---

## Short Comment (SC3) · 17 Oct 2017

Comments by S. A. Linge Johnsen and J. Bollmann

This paper is a valuable contribution to addressing the challenges of accurately estimating the weight of coccoliths using their birefringence. It confirms and validates the approach of Bollmann (2013a, 2014) of using material of known retardation for the calibration of the relationship between grey values and the thickness of coccoliths. Furthermore, it addresses some important shortcomings with previous attempts using rhabdoliths for calibration and the calculations of calcite thickness $>\sim1.5\mu$m. However, there are a few questions and comments that might improve the manuscript.

1) In the present study two polymer films were used to validate the relationship be-

tween grey values and retardation. How does this approach differ from that of Bollmann (2013a, 2014) who also used two polymer films to validate the relationship between grey values and retardation? Furthermore, Bollmann (2014, p.1908, recommendation #9) first suggested the construction of an empirically derived calibration curve using polymer sheets in steps of 20nm retardation and O'Dea et al. (2014) subsequently used this concept to calibrate the grey values and retardation relationship along equidistances on a quartz wedge. As the present manuscript basically uses the same approach as Bollmann (2013a, 2014) and O'Dea et al. (2014), it would be important to correctly acknowledge these contributions and to point out what the potential differences are (Pros and Cons). Furthermore, it would be useful to mention why a calcite wedge was used instead of a readily available quartz wedge.

2) Lochte (2014) already demonstrated the issues with using rhabdoliths for calibration while evaluating the calibration procedure described in Beaufort et al. (2014). Wouldn't it be appropriate to acknowledge and discuss her work? Furthermore, a recent study by Van De Locht et al. (2014) found a potentially hollow space within the spines of Rhabdosphaera clavigera using electron tomography. This would invalidate the assumption that rhabdolith thickness is equal to its width.

3) In light of the demonstrated issues with calibrating using a rhabdolith, how should data already published based on this methodology be interpreted?

4) The manuscript does not provide any statements/calculations about the accuracy and precision of the method. What is the accuracy and precision of the Zeiss Tilting Compensator B, and is it consistent over the entire wavelength/thickness? In Figure 2E the grey value minimum, indicating the position at which the calcite wedge reaches a given thickness, seems to span a distance from $\sim$8$\mu$m to $\sim$16$\mu$m for 1$\mu$m thickness on the calcite wedge. What is the justification for selecting a distance of $\sim$12.5$\mu$m for 1$\mu$m thickness? How would a distance of 8$\mu$m or 16$\mu$m, respectively, affect the calibration (see Figure 1 included with these comments)? The band of gray value minima for 2$\mu$m, 3$\mu$m and 4$\mu$m is much thinner than for 1$\mu$m and therefore either indicate that the accuracy of the Tilting Compensator B varies significantly with the thickness/retardation or that from ~8μm to ~16μm the thickness of the Calcite wedge is 1μm. Decreased accuracy at different retardations have been reported for various types of compensators before (e.g. Montarou, 2005; Sclar and Dillinger, 1960). Furthermore, the grey value curve is very noisy and was probably averaged. What is the standard deviation of a mean grey value at a given retardation/calibration point (e.g. on Figure 5A)?

5) Additionally, how may the optical resolution of the microscope and its calibration affect distance measurements along the wedge? How were the length measurements calibrated? What is the accuracy?

6) What are the associated uncertainties with the colour equations presented on table III? For example, what are the R2 and p values and how do uncertainties associated with the thickness measurements affect the accuracy and precision of the method? A complete discussion of uncertainties in measurements associated with the presented calibration method and how these affect the final coccolith thickness and mass measurements would be of great value, especially for the interpretation of weight trends and the comparison of data obtained with other methods.

7) The manuscript states several times that there is a theoretical sigmoidal relationship between grey values and thickness (e.g. on page 3 line 5-6, p. 5 l. 22-23). This statement is misleading and requires additional information about the digital image capturing. A sigmoidal shape of the grey value curve from 0 -~266nm can ONLY be obtained when a gamma of 1 was applied to an RGB image (e.g. RAW RGB image format) or when a Black and White camera was used that does not have RGB filters on the sensor (e.g. a Bayer Array). A gamma of 2.2 is required for images in sRGB or Adobe RGB (1998) colour space and it is usually automatically applied by the camera. Figure 2 (included with this comment) shows Michel-Lévy charts rendered with either gamma 1.0 (no gamma applied) or gamma 2.2 (standard for common RGB color spaces such as sRGB or Adobe RGB (1998)) that is converted into grey value curves. While no gamma shows a sigmoidal shaped curve, the 2.2 gamma chart shows a curve

similar to a Quadratic function (see also Bollmann, 2013b,c). The gamma applied to images and image formats should therefore be explicitly stated. Were images captured in RAW format and then converted into TIFF, JPEG etc.? If so, which algorithms were used? Which RGB colour space was used and was gamma applied to the images?

8) Further towards point 7: Page 6, line 7-9, quote: "Recent updating of the Michel-Levy curve (Sørensen, 2013) suggests that in the first order interference range the grayscale thickness relationship is better represented with a sigmoidal curve, an approach adopted by recent coccolith thickness studies (Beaufort et al., 2014;O'Dea et al., 2014)."

This statement is not correct. The revised Michel-Lévy chart by Sørensen (2013) does not show a sigmoidal relationship between grey values and retardation when interference colours represented in Adobe RGB (1998) colour space are converted into grey values. Sørensen (2013) revised ONLY the digital colour representation of the Michel-Lévy chart using transformations to reproduce the actual colour captured by digital cameras, including the transformation of light intensity into XYZ and RGB values and the application of gamma (see figure 2 in Sørensen (2013)). He did not revise the equations that describe the light transmission of the visible light spectrum with increasing retardation that was defined a long time ago based on equations by Fresnel (Fresnel, 1866; Johannsen, 1918). Only a gamma of 1 applied to an RGB image or the sum of the intensity of all wavelengths at a given retardation shows a sigmoidal curve with increasing retardation from 0- $\sim$ 266nm apparently referred to in the quote above. The latter can not be measured with a RGB camera (see also Bollmann 2013b,c)!

9) There are a few unclear points regarding the colour equations and thresholds used to define thickness in this study. First, referring to Figure 7A, threshold limits for V in Case 1 are set to V<130 or V<170, yet V increase above 170 for much of Case 1. Case 1 is furthermore defined slightly differently in Table II: "(110<H<160 & (S<80 or V<170)) or V<130". The "or" condition between S<80 and V<170 is not given in Figure 7A; which definition is the correct threshold? For Case 2, why is the threshold limit

for V set as low as >120 when in Figure 7A V is always >200 in the region defined by Case 2? Could Case 1 not be more easily defined by the previously described grey value relationship between 0 and 1.55$\mu$m? Lastly, Figure 7C could be improved by showing the different calibration points along the wedge where the colour equations were measured.

10) Page 2, Line 20: Wouldn't it be appropriate to include Craig (1961) in this list as he first described the employed technique for achieving circular polarization?

11) Page 3, Line 4-12. Why is the early approach of Beaufort (2005) not described in the summary of previous work to measure coccolith thickness from grey values? The method is obviously flawed in several ways and significantly differs from Bollmann's (2014, 2013a) approach of using polymers of know retardation (or any material of known retardation). However, Beaufort (2005) was first to use smear slides with a known weight of calcite particles to construct an empirical grey value calibration curve and this method has been used in several important studies.

12) The manuscript states several times that the grey value at saturation is 256 (e.g. Page 5: Line 9 and Page 9: Line 8). However, it should be 255, which is the maximum grey value in 8-bit images (0 =BLACK, 255 = WHITE in an 8-bit image with 256 grey values).

13) The readability of Table I could be improved by a more descriptive caption.

14) Why do the grey values in Figure 2B-D not reach lower than 70 (e.g. close to 0) at maximum extinction?

15) Figure 4: One of the R1 polymer film points seems to fall below the line obtained from the tilting compensator. Exactly how far from the tilting compensator line is the polymer point, and what could be the source of this deviation?

16) Figure 5A: The paper states that light saturation should be reached at 1.55$\mu$m, yet in Figure 5A, the calibration curve gives a grey value of approximately 200 at 1.5$\mu$m,

well below light saturation in 8-bit (255=WHITE). Why are the calibrated curves in Figure 5A not calibrated for light saturation at 1.55$\mu$m? Similarly, the calibration curve in Figure 6B seems to stop well short of light saturation at 1.55$\mu$m.

17) Figure 6B: Some rhabdoliths seem to have grey values which are much too high for their width/thickness. What could be the explanation for this? It seems that either these rhabdoliths are much thicker than they are wide or the images are overexposed.

18) Lastly, regarding Page 8, Line 18-24: "We suggest that several factors may cause variation in the color components for a given thickness. First, the spectrum of the microscope light source will vary the intensity at different color wavelengths and this may vary both among microscopes and over time due to bulb aging. Secondly, the use of filters, as well as objective characteristics, diaphragm aperture, light intensity, and light absorption by slides within the microscope system may affect the color components for a given thickness. Finally, within the digital camera, the quantum efficiency for a given wavelength may be different for different camera detectors."

Another major source of variation is the color transformation done by different color cameras, which should be corrected to decrease variations between different microscope setups (for details see Linge Johnsen et al., 2017).

References:

Beaufort, L., 2005. Weight estimates of coccoliths using the optical properties (birefringence) of calcite. Micropaleontology 51, 289–298. doi: 10.2113/gsmicropal.51.4.289

Beaufort, L., Barbarin, N., Gally, Y., 2014. Optical measurements to determine the thickness of calcite crystals and the mass of thin carbonate particles such as coccoliths. Nat. Protoc. 9, 633–42. doi:10.1038/nprot.2014.028

Bollmann, J., 2014. Technical Note: Weight approximation of coccoliths using a circular polarizer and interference colour derived retardation estimates - (The CPR Method). Biogeosciences 11, 1899–1910. doi:10.5194/bg-11-1899-2014

Bollmann, J., 2013a. Technical Note: Weight approximation of coccoliths using a circular polarizer and interference colour derived retardation estimates – (The CPR Method). Biogeosciences Discuss. 10, 11155–11179. doi:10.5194/bg-11-1899-2014

Bollmann, J. 2013b. Interactive comment on Biogeosciences Discuss., 10, 11155, C6879–C6886, 2013.

Bollmann, J. 2013c. Interactive comment on Biogeosciences Discuss., 10, 11155, C6961–C6981, 2013.

Charles B. Sclar; Lee Dillinger, 1960. The Microscopic Determination of The Thickness and Planeness of Platelets in Fine Materials. Am. Mineral. 45, 862–870.

Fresnel, A.-J., 1866. Ouevres complétes. Publiées par Henri de Sénarmont, Émile Verdet et Léonor Fresnel. Paris Impr. impériale 1866-1870, Paris, France.

González-Lemos, S., Guitián, J., Fuertes, M.-Á., Flores, J.-A., Stoll, H.M., 2017. An empirical method for absolute calibration of coccolith thickness. Biogeosciences Discuss. doi:https://doi.org/10.5194/bg-2017-249

Johannsen, A., 1918. Manual of petrographic methods, 2nd ed. McGraw-Hill, New York, NY, USA.

Linge Johnsen, S.A., Bollmann, J., Lee, H.W., Zhou, Y., 2017. Accurate representation of interference colours (Michel-Lévy chart): from rendering to image colour correction. J. Microsc. doi:10.1111/jmi.12641

Lochte, A.A., 2014. Single Coccolith Weight Estimates on Cultured Gephyrocapsa oceanica. Master's thesis. Uppsala University. Retrieved from: http://uu.diva-portal.org/smash/record.jsf?pid=diva2%3A739268&dswid=6623

Montarou, C.C., 2005. Low-Level Birefringence Measurement Methods Applied to The Characterization of Optical Fibers and Interconnects. Phd thesis. Georgia Institute of Technology. Retrieved at: https://smartech.gatech.edu/handle/1853/6993

O'Dea, S.A., Gibbs, S.J., Bown, P.R., Young, J.R., Poulton, A.J., Newsam, C., Wilson, P. a., 2014. Coccolithophore calcification response to past ocean acidification and climate change. Nat. Commun. 5, 5363. doi:10.1038/ncomms6363

Sørensen, B.E., 2013. A revised Michel-Lévy interference colour chart based on first-principles calculations. Eur. J. Mineral. 25, 5–10. doi:10.1127/0935-1221/2013/0025-2252

Van De Locht, R., Slater, T.J.A., Verch, A., Young, J.R., Haigh, S.J., Kröger, R., 2014. Ultrastructure and crystallography of nanoscale calcite building blocks in Rhabdosphaera clavigera coccolith spines. Cryst. Growth Des. 14, 1710–1718. doi:10.1021/cg4018486
* * *
[Figure]

E

| 0 µm | 1 µm | 2 µm | 3 µm | 4 µm |

**Fig. 1.** Figure 2E from González-Lemos et al. (2017) with minimum area for $1\mu$m thickness of calcite wedge highlighted by the pink box. The black arrow in the pink box shows the distance position of $\sim12.5\mu$m on

**A**

**B**

**C**

Gray value

Retardation (nm, above) and thickness (μm, below)

Gamma 1.0  Gamma 2.2

**Fig. 2.** The relationship between retardation/thickness and gray values of two Michel-Lévy charts in the 0-688nm retardation/0-4$\mu$m calcite thickness range produced in sRGB color space with 3200K color temperat

---

## Author Comment (AC2) · 23 Nov 2017

The reviewer mentiones the idea of employing diferences in wavelength when narrow band color filters are using for illumination, as an alternative approach for estimating differential retardation and thickness. This is an interesting idea, for which we are not aware of any previous work, and could be fruitful direction for future study.

The reviewer comments that it would be useful to estimate the distribution of C axis orientation in coccoliths. The orientation of the crystallographic c-axis has been reviewed thoroughly by Young et al., 2004 : Young, J.R., Herniksen, K., and Probert, I.: Structure and morphogenesis of the coccoliths of the CODENET species. In: Trierstein, H.R., Young, J.R.: Coccolithophores: from molecular processes to global impact; Ed.:

(eds.), Springer-Verlag Berlin Heidelberg. 2014. We have carried out analysis with an electron microscope only to evaluate the preservation state of the sampled coccoliths.

Finally, we thank the reviewer for suggestions on fixing some minor typographical errors, which could be attended to in revised version of the manuscript.

---

## Author Comment (AC3) · 23 Nov 2017

The reviewer asks about evidence for the preservation status of the Rhabdosphera employed in the measurements which have given inverse relationships, and the subtraction of the background.

There are no evidences of poor preservation of Rhabdosphaeras (R9 and R10). Of course, we are agree that these specimens should not be considered for calibration. We have represented these specimens on the graph to show the high range of variation existing according to the Rhabdosphera chosen, since in most cases only a single specimen is used to calibrate a measurement series.

We clarify that the background gray level of sample slides is subtracted from all images

before quantifying gray level of coccoliths or nannoliths. A more detailed description of the process would be "After subtracting the background gray level from the image, for each rhabdolith we made 10 measurements of width/thickness and its corresponding gray level at different points." Consequently, the Rhabdosphaera width is defined by all pixels with a grayscale value greater than 0.
* * *

---

## Author Comment (AC4) · 23 Nov 2017

We clarify that, unlike the Bollmann (2013a, 2014), the present study uses two polymer films to only validate two real thickness points on the calcite wedge. The calcite wedge is used to provide a continuous calibration material over the thickness range from 0 to 4 microns, including color range. This contrasts with the previous approaches and recommendations of Bollmann (2014) which used and recommended multi-polymers to establish a multi-point calibration between only grayscale and thickness (without entering color range).

We clarify that a calcite wedge was used because it permits direct comparison between interference colors in the wedge and in the coccoliths, because both are made

of calcite with the same birefringence. Using a quartz wedge would require adjustment for the different birefringence of quartz. We appreciate the reviewer alerting us to the unpublished Masters thesis of Lochte et al as an initial mention of challenge with Rhabdospheaera calibration. The findings in this thesis are coherent with ours, that there are challenges with Rhabdosphaera calibration. Also, the reviewers suggestion as a potential explanation for the divergent results of Rhabdosphaera, the proposal by Van de Locht et al. (2014) that Rhabdosphaera spines may have a hollow space, is worthy of inclusion.

The reviewer queries what conclusion can be made from previous studies using Rhabdosphera calibration. As we had stated in the manuscript, calibrations with a single rhabdolith will produce data which is internally consistent (e.g. relative trends will be robust), but the absolute thickness measurements may not be comparable. In general, we suggest that an inter-laboratory calibration exercise is needed because of the diversity of calibration approaches previously employed, to ensure that data generated in the future in different laboratories can be compared with confidence.

Regarding the accuracy of the Zeiss tilting compensator, the measuring accuracy of the magnesium fluoride tilting compensator given by the manufacturer is $\pm \sim$2.5-8 nm of change in optical path difference ($\Delta$OPD). The precision decreases, indeed, with increasing tilting angle, but the advantage of the rotating compensator is that they have more constant accuracy for all the positions of the compensator crystal.

We can clarify that small variations in the slope of the calcite wedge give rise to different widths of the gray band shown in Figure 2E. In particular, in figure 2E, the first zone features a very low slope on the calcite wedge, leading to a wider band of gray with the tilting compensator, and the value of 12.5 $\mu$m represents the midpoint. They gray value curve in Figure 2E was not averaged therefore there is no standard deviation to report. Improving the manufacturing of following calcite wedges would avoid this situation. All length measurements are held to the Zeiss Axiocam camera resolution, therefore 1 pixel = 0.0454 microns.

To clarify the uncertainties in the color equations, we now list the R2 and p-values of each individual component of the regression lines in Table III (appended). For the overall application, we provide an estimate of uncertainty by reserving the majority of the pixels along the profile for validation and only using a small fraction for calibration. We now report the R2 and p-value for this validation relationship shown in Figure 7D (appended).

We clarify that a gamma of 1 correction value was applied on the 3 channels (RGB), which is consistent with the sigmoidal shape obtained between gray values and thickness. No filter, color correction or similar was applied during image taking. The images were saved as tiff format without any kind of compression. The camera and acquisition software use standard RGB color space (sRGB). Since a gamma 1.0 value was used when images were acquired and saved as TIFF format, the sigmoidal shape of the grey values curve from 0 – 266 nm was obtained.

We have corrected the thresholds and equations reported in Figure 7A (appended), which did not match those of Table III and the text; the values reported in Table III and the text were correct. Furthermore, in case 2, the value of 120 for V is set at a low value to avoid erroneously excluding pixels which are into this thickness range but where the V value is not reaching the maximum. Finally, for case 1, we prefer to work already with RGB values also for this range, even if greyscale conditions could be established as described. In that way we have all the data points with the same code and we avoid conversion errors while running the script.

Regarding the introduction, we agree it is useful to detail the early calibration approach, that "Beaufort (2005) was first to use smear slides with a known weight of calcite particles to construct an empirical grey value calibration curve, whereas O'Dea (2014) applied a theoretical sigmoidal relationship between grayscale and thickness.

We agree that refinement is needed in the definition of saturation, when we are talking about the light saturation is defined as sum of RGB as 256 levels of gray. When we

refer to the saturation limit elsewhere we describe the limit (white = 255).

Likewise Table I legend should specify more clearly, Calibration values for 15 individual Rhabdosphaera clavigera rods (R1 to R15) measured under identical light conditions. Sigmoidal and linear fits are shown. The lowest deviation is achieved when sigmoidal calibration is applied.

The reviewer questions the minimum values in Figure 2. First of all, the original panels B-D were taken simply to illustrate schematically the procedure but were from a different camera and microscope system than the one used to calibrate the wedge. We have now updated the figure (appended) to illustrate the individual curves shown previously in Panel E, those actually used to carry out the calibration. In this case, it is possible to see that the 0 $\mu$m compensation grey level curve has a minimum of 25, which is close to the background level for the microscope/camera setup, and which may be slightly elevated if the exact boundary of the wedge does not have a perfect taper to 0 $\mu$m thickness. Likewise we have corrected the Figure 4 (appended) to evaluate the previous divergence for the first polymer. All images have been checked and retaken in order to evaluate this divergence. When the saturation is not reached properly for the polymer images with the microscope software, the saturation on the wedge would be attained at slightly different positions.

We clarify that for work limited to grayscale range, the light intensity was adjusted to give an optimal range of grayscale values for the particles of interest. In contrast, for work including the color range, the light intensity was adjusted to attain saturation at a calcite thickness of 1.55 ïA■m." For this reason, the grayscale range images in Figure 5A did not correspond to saturation conditions at 1.55 $\mu$m. The purpose of Figure 5A is to show the consistency and reproducibility of the calcite wedge on images taken under similar microscope settings (and light intensity) on 10/20/2014 and 07/15/2015.

Figure 6B (appended): We have corrected an error in the plot of the calcite wedge curve, which mistakenly applied a calcite wedge image taken under different microscope settings than the Rhabdosphaera rods. Now, the calibration curve corresponds to the same microscope settings used for Rhabdosphaera specimens. The light saturation (Gray Level = 255) is reached at 1.55 $\mu$m. We clarify that all images of Rhabdosphaeras are well focused and are not overexposed. A possible explanation is that the rabdoliths may not be perfectly cylindrical, so that the width does not correspond with the thickness assigned from gray level.

We agree that it is useful to acknowledge, as the reviewer suggested, that another major source of variation is the color transformation done by different color cameras, which should be corrected to decrease variations between different microscope setups (Johnsen et al., 2017)..."
* * *
Table III. Definition of four unique or combination color components and the relevant equations for the thickness calculation for our microscope conditions. The calibration was established from the calcite wedge ETH-W2. P-values for the regression are all below 0.01.

| | Thickness Range | Thresholds | Thickness proportional to | Equation | $R^2$ |
|---|---|---|---|---|---|
| Case 1 | 0 – 1.4 μm | (110<H<160 & (S<80 or V<170)) or V<130 | Rising V | $T = 8.806E\text{-}10x^4 - 4.086E\text{-}07x^3 + 7.336E\text{-}05x^2 - 1.382E\text{-}03x + 7.080E\text{-}02$ | 0.9699 |
| Case 2 | 1.4 – 2.5 μm | 19<H<120 & S<200 & V>120 | Rising S-V | $T = -1.171E\text{-}08x^4 - 7.856E\text{-}06x^3 - 1.961E\text{-}03x^2 - 2.102E\text{-}01x - 5.722E\text{+}00$ | 0.9298 |
| Case 3 | 2.5 – 3 μm | H<19 & S>0 & V>150 | Decreasing V | $T = 2.318E\text{-}07x^4 - 1.930E\text{-}04x^3 + 6.000E\text{-}02x^2 - 8.260E\text{+}00x + 4.281E\text{+}02$ | 0.9613 |
| Case 4 | 3 – 4 μm | ELSE | Decreasing H | $T = 2.164E\text{-}08x^4 - 1.881E\text{-}05x^3 + 6.034E\text{-}03x^2 - 8.527E\text{-}01x + 4.842E\text{+}01$ | 0.9460 |

**Fig. 1.** Table III

**Fig. 2.** Figure 2 revised

A

**Gray Level** (y-axis, values: 0, 50, 100, 150, 200, 250)

Legend: R1, R2, R3, R4, R5, R6, R7, R8, R9, R10, R11, R12, R13, R14, R15

**Rhabdolith width/thickness (μm)** (x-axis: 0, 0.2, 0.4, 0.6, 0.8, 1, 1.2, 1.4)

B

**Gray Level** (y-axis, values: 0, 50, 100, 150, 200, 250)

Legend: R1, R2, R3, R4, R5, R6, R7, R8, R9, R10, R11, R12, R13, R14, R15, calcite wedge

**Rhabdolith width/thickness (μm)** (x-axis: 0, 0.2, 0.4, 0.6, 0.8, 1, 1.2, 1.4)

**Fig. 3.** Figure 6 revised

[Figure]

**Fig. 4.** Figure 4 revised

[Figure]

**Fig. 5.** Figure 7 revised

---

## Author Response (AR1)

**Point by point response to reviewers of manuscript entitled** *An empirical method for absolute calibration of coccolith thickness* **(DOI: 10.5194/bg-2017-249-SC2).**

The reviewers comments are in **blue** and our response in **black**.

5  *Interactive comment of X. Jin.

Page 4, line 12: "one is located between the lower linear polarizer and the condenser and the other one above the upper linear polarizer" the other one should be below the upper linear polarizer.

We have corrected this typographic error in the text. The microscope set up was indeed correct. Therefore, as you say, the other $\lambda/4$ plate should be below the upper linear polarizer, and not above it.

10  *Interactive comment of Anonymous Referee #1.

This is a very well written paper that clearly sets out all the steps in obtaining an accurate calibration of thickness for coccoliths using polarization techniques in an imaging microscope. I found the paper easy to follow even for someone coming more from the optical physics side of the house. Every step is clearly described, all the sources of error are properly addressed and the procedures required to adapt other equipment and duplicate the fabrication of the wedge standard are clearly outlined ensuring

15  that this paper will significant and of use to many other researchers in the field. I therefore recommend its publication. I only found one minor typo which is a testimony to the very good quality of the authors writing: Page 6 Line19: "that they rods are cylindrical" should read "that they are cylindrical rods"

We have incorporated this change in the new version of the manuscript.

The following comments are queries which come from my perspective as an optical physicist and should not in any way be

20  interpreted as a criticism of the paper. They may mostly be based on my ignorance of the customs and literature in the field. It seems to me that first glance that a relatively simple model for the polarization interference pattern as a function of illumination wavelength (even in the color range) of a given thickness of calcite could be derived as the retardation dispersion has been well measured. [G. Ghosh. Dispersion-equation coefficients for the refractive index and birefringence of calcite and quartz crystals, Opt. Commun. 163, 95-102 (1999)]. In and of itself, this may not be precise enough for a calibration but it would be

25  precious to have as a guide to the proper functional form of a calibration curve. As anyone considered using a well-chosen set of narrow band color filters in front of the illumination source of the microscope? The intensity of each wavelength for a given pixel could be recorded and the ratios calibrated against a blank slide response computed. These would give information on the differential retardation and therefore the thickness given the known indices of refraction of calcite. If a black and white camera where used this would be completely straightforward. If a color camera where used the camera color channel response

matrix could be used for the same purpose.

This is an interesting idea which we have not tried, but could be fruitful for a future work.

Finally, it seems to me that comparing the retardation map with an electron microscope size map could in principle be used to estimate the distribution of the c-axis crystal orientation in the lith structure. Has this been done by anyone?

5  The distribution of c-axis crystal orientation in coccolith has been reviewed thoroughly by Young et al., 2004. We have carried out analysis with an electron microscope only to evaluate the preservation state of the sampled coccoliths.

*Interactive comment of X. Jin (Referee – 386jinxiaobo@tongji.edu.cn).

This paper is concerning about a calibration between coccoliths thickness and their optical properties under circular polarized light microscope. And the usage of calcite wedges for the calibration provides a possibility and also a criteria to compare
10  coccolith thickness and weight as estimated by their optical features when using light microscopes and cameras with different settings in other labs. The paper is well written, and authors provide a detailed technological processes for calcite wedges manufacture, and for measurements of the thickness of calcite wedges. So I see this study is important and recommend it can be published in Biogeosciences. Still, I have several questions that listed below: I have found inverse relations between grey level and width of Rhabdosphaera species (R9, R10) in figure 6A. Are these specimens poorly preserved? These specimens
15  may not be considered for calibration.

There are no evidences of poor preservation of *Rhabdosphaeras* (R9 and R10). Of course, we are agree that these specimens should not be considered for calibration. We have represented these specimens on the graph to show the high range of variation existing according to the *Rhabdosphera* chosen, since in most cases only a single specimen is used to calibrate a measurements series.

20  It can be found that the linear K values of relation between grey level and width of Rhabdosphaera increase with the incensing of their width, when the width is <0.6 µm most Rhabdosphaera measurements are below the calcite wedge curve, and when >0.6µm most Rhabdosphaera are above the calcite wedge curve. So compared with the calcite wedge, all the Rhabdosphaera measurements make their calibration be more like a sigmoidal function. Is the background grey levels of the coccolith sample slides considered in the Rhabdosphaera calibration? And how about the grey level threshold for Rhabdosphaera coccoliths
25  outline constrain? This is important for calibration, since for lighter/brighter part of Rhabdosphaera robs, their width could be overestimated due to dispersion.

We have clarified in the text that the background gray level of sample slides is subtracted from all images before quantifying gray level of coccoliths or nannoliths. Page 6: Lines 29-31: *"After subtracting the background gray level from the image, and for each rhabdolith we made 10 measurements of width/thickness and its corresponding gray level at different points."*
30  Consequently, the *Rhabdosphaera* width is defined by all pixels with a grayscale value greater than 0.

Technical corrections: Page 4 line 12: the other one "above"… below

It has been corrected in line 16: page 4 of the new version of the manuscript.

*Interactive comment of S. A. Linge Johnsen (simen.johnsen@mail.utoronto.ca) and J. Bollmann.

This paper is a valuable contribution to addressing the challenges of accurately estimating the weight of coccoliths using their birefringence. It confirms and validates the approach of Bollmann (2013a, 2014) of using material of known retardation for the calibration of the relationship between grey values and the thickness of coccoliths. Furthermore, it addresses some important shortcomings with previous attempts using rhabdoliths for calibration and the calculations of calcite thickness >1.5µm. However, there are a few questions and comments that might improve the manuscript.

1) In the present study two polymer films were used to validate the relationship between grey values and retardation. How does this approach differ from that of Bollmann (2013a, 2014) who also used two polymer films to validate the relationship between grey values and retardation? Furthermore, Bollmann (2014, p.1908, recommendation #9) first suggested the construction of an empirically derived calibration curve using polymer sheets in steps of 20nm retardation and O'Dea et al. (2014) subsequently used this concept to calibrate the grey values and retardation relationship along equidistances on a quartz wedge. As the present manuscript basically uses the same approach as Bollmann (2013a, 2014) and O'Dea et al. (2014), it would be important to correctly acknowledge these contributions and to point out what the potential differences are (Pros and Cons). Furthermore, it would be useful to mention why a calcite wedge was used instead of a readily available quartz wedge.

We clarify that, unlike the Bollmann (2013a, 2014), the present study uses two polymer films to only validate two real thickness points on the calcite wedge. The calcite wedge is used to provide a continuous calibration material over the thickness range from 0 to 4 microns, including color range. This contrasts with the previous approaches and recommendations of Bollmann (2014) which used and recommended multi-polymers to establish a multi-point calibration between only grayscale and thickness (without entering color range).

We make clear now at the end of the introduction, that a calcite wedge was used because it permits direct comparison between interference colors in the wedge and in the coccoliths, because both are made of calcite with the same birefringence. Using a quartz wedge would require adjustment for the different birefringence of quartz. We have added at Page 2: Lines 23-24: "*A calcite wedge was used because it permits direct comparison between interference colors in the wedge and in the coccoliths, because both are made of calcite with the same birefringence*".

2) Lochte (2014) already demonstrated the issues with using rhabdoliths for calibration while evaluating the calibration procedure described in Beaufort et al. (2014). Wouldn't it be appropriate to acknowledge and discuss her work? Furthermore, a recent study by Van De Locht et al. (2014) found a potentially hollow space within the spines of Rhabdosphaera clavigera using electron tomography. This would invalidate the assumption that rhabdolith thickness is equal to its width.

We now mention the Lochte et al. reference (an otherwise unpublished master's thesis) as an initial mention of challenge with *Rhabdospheaera* calibration. We have added *"Lochte (2014) tested two calibration techniques proposed by Beaufort et al. (2014) and Bollmann (2014) to compare coccolith weight estimates of cultured single clones of Gephyrocapsa oceanica, showing of challenge with Rhabdosphaera calibration"* in the new version of the manuscript (Page 3: Lines 12 – 14).

5   We have added as a potential explanation for the divergent results of *Rhabdosphaera*, the proposal by Van de Locht et al. (2014) that *Rhabdosphaera* spines may have a hollow space. We have modified the Page 7: Lines 11-13 in the new version of the manuscript: *"Van de Locht et al. (2014) found, using electron tomography, that Rhabdosphaera rods may have a hollow space. This morphology could provide an explanation for the divergent results of Rhabdosphaera"*.

3) In light of the demonstrated issues with calibrating using a rhabdolith, how should data already published based on this
10  methodology be interpreted?

As we had stated in the manuscript that calibrations with a single rhabdolith will produce data which is internally consistent (e.g. relative trends will be robust), but the absolute thickness measurements may not be comparable. In general, we suggest that an inter-laboratory calibration exercise is needed because of the diversity of calibration approaches previously employed, to ensure that data generated in the future in different laboratories can be compared with confidence.

15  4) The manuscript does not provide any statements/calculations about the accuracy and precision of the method. What is the accuracy and precision of the Zeiss Tilting Compensator B, and is it consistent over the entire wavelength/thickness? In Figure 2E the grey value minimum, indicating the position at which the calcite wedge reaches a given thickness, seems to span a distance from 8μm to 16 µm for 1 µm thickness on the calcite wedge. What is the justification for selecting a distance of 12.5 µm for 1 µm thickness? How would a distance of 8 µm or 16 µm, respectively, affect the calibration (see Figure 1 included
20  with these comments)? The band of gray value minima for 2 µm, 3 µm and 4 µm is much thinner than for 1 µm and therefore either indicate that the accuracy of the Tilting Compensator B varies significantly with the thickness/retardation or that from 8 µm to 16 µm the thickness of the Calcite wedge is 1 µm. Decreased accuracy at different retardations have been reported for various types of compensators before (e.g. Montarou, 2005; Sclar and Dillinger, 1960). Furthermore, the grey value curve is very noisy and was probably averaged. What is the standard deviation of a mean grey value at a given retardation/calibration
25  point (e.g. on Figure 5A)?

The measuring accuracy of the magnesium fluoride tilting compensator given by the manufacturer is ± ~2.5-8 nm of change in optical path difference (ΔOPD). The precision decreases, indeed, with increasing tilting angle, but the advantage of the rotating compensator is that they have more constant accuracy for all the positions of the compensator crystal. We have added to the new version of the manuscript Page 4: Line 30: *"The reported accuracy of the instrument is ± ~2.5-8 nm of variation in*
30  *optical path difference (ΔOPD)."*

We have clarified in the manuscript that small variations in the slope of the calcite wedge give rise to different widths of the gray band shown in Figure 2E. We have modified Page 5: Lines 6-7: *"Small variations in the slope of the calcite profile cause*

*different widths of the dark full compensation zone.*"

In figure 2E, the first zone features a very low slope on the calcite wedge, leading to a wider band of gray with the tilting compensator, and the value of 12.5 µm represents the midpoint. They gray value curve in Figure 2E was not averaged therefore there is no standard deviation to report. Improving the manufacturing of following calcite wedges would avoid this situation.

5) Additionally, how may the optical resolution of the microscope and its calibration affect distance measurements along the wedge? How were the length measurements calibrated? What is the accuracy?

All length measurements are held to the Zeiss Axiocam camera resolution, therefore 1 pixel = 0.0454 microns.

6) What are the associated uncertainties with the colour equations presented on table III? For example, what are the R2 and p values and how do uncertainties associated with the thickness measurements affect the accuracy and precision of the method? A complete discussion of uncertainties in measurements associated with the presented calibration method and how these affect the final coccolith thickness and mass measurements would be of great value, especially for the interpretation of weight trends and the comparison of data obtained with other methods.

We now list the R2 and p-values of each individual component of the regression lines in Table III. For the overall application, we provide an estimate of uncertainty by reserving the majority of the pixels along the profile for validation and only using a small fraction for calibration. We now report the R2 and p-value for this validation relationship shown in Figure 7D.

7) The manuscript states several times that there is a theoretical sigmoidal relationship between grey values and thickness (e.g. on page 3 line 5-6, p. 5 l. 22-23). This statement is misleading and requires additional information about the digital image capturing. A sigmoidal shape of the grey value curve from 0 -266nm can ONLY be obtained when a gamma of 1 was applied to an RGB image (e.g. RAW RGB image format) or when a Black and White camera was used that does not have RGB filters on the sensor (e.g. a Bayer Array). A gamma of 2.2 is required for images in sRGB or Adobe RGB (1998) colour space and it is usually automatically applied by the camera. Figure 2 (included with this comment) shows Michel-Lévy charts rendered with either gamma 1.0 (no gamma applied) or gamma 2.2 (standard for common RGB color spaces such as sRGB or Adobe RGB (1998)) that is converted into grey value curves. While no gamma shows a sigmoidal shaped curve, the 2.2 gamma chart shows a curvesimilar to a Quadratic function (see also Bollmann, 2013b,c). The gamma applied to images and image formats should therefore be explicitly stated. Were images captured in RAW format and then converted into TIFF, JPEG etc.? If so, which algorithms were used? Which RGB colour space was used and was gamma applied to the images?

A gamma of 1 correction value was applied on the 3 channels (RGB), which is consistent with the sigmoidal shape obtained between gray values and thickness. No filter, color correction or similar was applied during image taking. The images were saved as tiff format without any kind of compression. The camera and acquisition software use standard RGB color space (sRGB).

8) Further towards point 7: Page 6, line 7-9, quote: "Recent updating of the Michel-Levy curve (Sørensen, 2013) suggests that

in the first order interference range the grayscale thickness relationship is better represented with a sigmoidal curve, an approach adopted by recent coccolith thickness studies (Beaufort et al., 2014;O'Dea et al., 2014)."

This statement is not correct. The revised Michel-Lévy chart by Sørensen (2013) does not show a sigmoidal relationship between grey values and retardation when interference colours represented in Adobe RGB (1998) colour space are converted into grey values. Sørensen (2013) revised ONLY the digital colour representation of the Michel-Lévy chart using transformations to reproduce the actual colour captured by digital cameras, including the transformation of light intensity into XYZ and RGB values and the application of gamma (see figure 2 in Sørensen (2013)). He did not revise the equations that describe the light transmission of the visible light spectrum with increasing retardation that was defined a long time ago based on equations by Fresnel (Fresnel, 1866; Johannsen, 1918). Only a gamma of 1 applied to an RGB image or the sum of the intensity of all wavelengths at a given retardation shows a sigmoidal curve with increasing retardation from 0- 266nm apparently referred to in the quote above. The latter can not be measured with a RGB camera (see also Bollmann 2013b,c))!

As we have responded in previous comment 7, a gamma 1.0 value was used when images were acquired and saved as TIFF format. Then the sigmoidal shape of the grey values curve from 0 – 266 nm was obtained.

9) There are a few unclear points regarding the colour equations and thresholds used to define thickness in this study. First, referring to Figure 7A, threshold limits for V in Case 1 are set to V<130 or V<170, yet V increase above 170 for much of Case 1. Case 1 is furthermore defined slightly differently in Table II: "(110<H<160 & (S<80 or V<170)) or V<130". The "or" condition between S<80 and V<170 is not given in Figure 7A; which definition is the correct threshold? For Case 2, why is the threshold limit for V set as low as >120 when in Figure 7A V is always >200 in the region defined by Case 2? Could Case 1 not be more easily defined by the previously described grey value relationship between 0 and 1.55 µm? Lastly, Figure 7C could be improved by showing the different calibration points along the wedge where the colour equations were measured.

We have corrected the thresholds and equations reported in Figure 7A, which did not match those of Table III and the text; the values reported in Table III and the text were correct. Furthermore, in case 2, the value of 120 for V is set at a low value to avoid erroneously excluding pixels which are into this thickness range but where the V value is not reaching the maximum. Finally, for case 1, we prefer to work already with RGB values also for this range, even if greyscale conditions could be established as described. In that way we have all the data points with the same code and we avoid conversion errors while running the script.

10) Page 2, Line 20: Wouldn't it be appropriate to include Craig (1961) in this list as he first described the employed technique for achieving circular polarization?

We now include this citation. Page 2: Lines 19-20: "*In this work, following previous studies (Bollmann, 2014;Craig, 1961;Fuertes et al., 2014)*".

11) Page 3, Line 4-12. Why is the early approach of Beaufort (2005) not described in the summary of previous work to measure coccolith thickness from grey values? The method is obviously flawed in several ways and significantly differs from

Bollmann's (2014, 2013a) approach of using polymers of know retardation (or any material of known retardation). However, Beaufort (2005) was first to use smear slides with a known weight of calcite particles to construct an empirical grey value calibration curve and this method has been used in several important studies.

This work is also cited in the introduction. We have modified the Lines 6-8: Page 3 in the new version of the manuscript in order to add this early calibration approach: *"Beaufort (2005) was first to use smear slides with a known weight of calcite particles to construct an empirical grey value calibration curve. O'Dea (2014) applied a theoretical sigmoidal relationship between grayscale and thickness.*

12) The manuscript states several times that the grey value at saturation is 256 (e.g. Page 5: Line 9 and Page 9: Line 8). However, it should be 255, which is the maximum grey value in 8-bit images (0 =BLACK, 255 = WHITE in an 8-bit image with 256 grey values).

Page 5, Line 9: we are talking about the light saturation is defined as sum of RGB as 256 levels of gray. Page 9, Line 21: We have corrected in the new version of the manuscript the value "256" to *"255"*, since in this case we are talking to the saturation limit (white = 255).

13) The readability of Table I could be improved by a more descriptive caption.

We now clarify the headline in table I in the new version of the manuscript: *"Calibration values for 15 individual Rhabdosphaera clavigera rods (R1 to R15) measured under identical light conditions. Sigmoidal and linear fits are shown. The lowest deviation is achieved when sigmoidal calibration is applied"*.

14) Why do the grey values in Figure 2B-D not reach lower than 70 (e.g. close to 0) at maximum extinction?

First of all, the original panels B-D were taken simply to illustrate schematically the procedure but were from a different camera and microscope system than the one used to calibrate the wedge. We have now updated the figure to illustrate the individual curves shown previously in Panel E, those actually used to carry out the calibration.
In this case, it is possible to see that the 0 µm compensation grey level curve has a minimum of 25, which is close to the background level for the microscope/camera setup, and which may be slightly elevated if the exact boundary of the wedge does not have a perfect taper to 0 µm thickness.

15) Figure 4: One of the R1 polymer film points seems to fall below the line obtained from the tilting compensator. Exactly how far from the tilting compensator line is the polymer point, and what could be the source of this deviation?

All images have been checked and retaken in order to evaluate this divergence. It has been corrected in figure 4 of the new version of the manuscript. When the saturation is not reached properly for the polymer images with the microscope software, the saturation on the wedge would be attained at slightly different positions.

16) Figure 5A: The paper states that light saturation should be reached at 1.55 µm, yet in Figure 5A, the calibration curve gives

a grey value of approximately 200 at 1.5 µm, well below light saturation in 8-bit (255=WHITE). Why are the calibrated curves in Figure 5A not calibrated for light saturation at 1.55 µm? Similarly, the calibration curve in Figure 6B seems to stop well short of light saturation at 1.55 µm.

We have clarified in the text that for work limited to grayscale range, the light intensity was adjusted to give an optimal range
5    of grayscale values for the particles of interest. We have modified Page 4: Lines 20-22: "*For work limited to grayscale range, the light intensity was adjusted to give an optimum range in grayscale values for the materials being quantified. For work including the color range, the light intensity was adjusted to attain saturation at a calcite thickness of 1.55 µm.*" and Page 6: Lines 1-3: "*We emphasize that for work limited to grayscale range, the light intensity was adjusted to give an optimum range in grayscale values for the materials being quantified*".

10    The images in Figure 5A did not correspond to saturation conditions at 1.55 µm. The purpose of Figure 5A is to show the consistency and reproducibility of the calcite wedge on images taken under similar microscope settings (and light intensity) on 10/20/2014 and 07/15/2015. We have also clarified on Figure 2 and Figure 7 text: "*Light intensity was saturated at a calcite thickness of 1.55 µm.*"

Figure 6B: We have corrected an error in the plot of the calcite wedge curve, which mistakenly applied a calcite wedge image
15    taken under different microscope settings than the *Rhabdosphaera* rods. Now, the calibration curve corresponds to the same microscope settings used for *Rhabdosphaera* specimens. The light saturation (Gray Level = 255) is reached at 1.55 µm.

17) Figure 6B: Some rhabdoliths seem to have grey values which are much too high for their width/thickness. What could be the explanation for this? It seems that either these rhabdoliths are much thicker than they are wide or the images are overexposed.

20    All images of *Rhabdosphaeras* are well focused and are not overexposed. A possible explanation is that the rabdoliths may not be perfectly cylindrical, so that the width does not correspond with the thickness assigned from gray level.

18) Lastly, regarding Page 8, Line 18-24: "We suggest that several factors may cause variation in the color components for a given thickness. First, the spectrum of the microscope light source will vary the intensity at different color wavelengths and this may vary both among microscopes and over time due to bulb aging. Secondly, the use of filters, as well as objective
25    characteristics, diaphragm aperture, light intensity, and light absorption by slides within the microscope system may affect the color components for a given thickness. Finally, within the digital camera, the quantum efficiency for a given wavelength may be different for different camera detectors."

Another major source of variation is the color transformation done by different color cameras, which should be corrected to decrease variations between different microscope setups (for details see Linge Johnsen et al., 2017).

30    We have modified the Line 34: Page 8 and Line 1: Page 9 in the new version of the manuscript in order to add this comment: "*Also, another major source of variation is the color transformation done by different color cameras, which should be*

*corrected to decrease variations between different microscope setups (Johnsen et al., 2017)...*"

[Figure]

Fig. 1. Figure 2E from González-Lemos et al. (2017) with minimum area for 1 µm thickness of calcite wedge highlighted by the pink box. The black arrow in the pink box shows the distance position of 12.5 µm on

[Figure]

Fig. 2. The relationship between retardation/thickness and gray values of two Michel-Lévy charts in the 0-688nm retardation/0-4 µm calcite thickness range produced in sRGB color space with 3200K color temperat

[revised manuscript text omitted]

---

## Author Response (AR2)

Dear Editor van der Meer

I upload the revised version of the manuscript. I have personally done a careful editing to reduce as much as possible the use of active voice (We did..) and employ more passive voice, and ensure that passive voice in the method development is all in the same tense (past). Description of results remains in present tense.

The word microns has been replaced everywhere with the symbol, and the correlation sign is now expressed verbally as positive

10 or negative rather than with use of a – sign for negative correlations. All wording has been clarified where suggested, including the addition of "the" where indicated.

In response to the suggestion to reduce the number of tables and figures, we have eliminated figure 1 and also Table 1. The range in calibration values previously tabulated in Table 1 is now summarized in the caption for Figure 5. We also evaluated eliminating the Former Table 2, however as many of the composite parameters (such as differences or sums of color values)

15 are not illustrated in Figure we consider it important to leave it in.

Below you will find the revised papers with track changes.

I hope you are satisfied with these revisions, and thank you for your attention to making the paper as clear and readable as possible.

20 Best regards
Heather Stoll

[revised manuscript text omitted]